# Fine Identification and Classification of a Novel Beneficial *Talaromyces* Fungal Species from Masson Pine Rhizosphere Soil

**DOI:** 10.3390/jof8020155

**Published:** 2022-02-03

**Authors:** Xiao-Rui Sun, Ming-Ye Xu, Wei-Liang Kong, Fei Wu, Yu Zhang, Xing-Li Xie, De-Wei Li, Xiao-Qin Wu

**Affiliations:** 1Co-Innovation Center for Sustainable Forestry in Southern China, College of Forestry, Nanjing Forestry University, Nanjing 210037, China; taylorrui240@njfu.edu.cn (X.-R.S.); xumingye1121@163.com (M.-Y.X.); k3170100077@njfu.edu.cn (W.-L.K.); wufeinjfu@sina.com (F.W.); songhongyusorry11@163.com (Y.Z.); XingLiXIE@njfu.edu.cn (X.-L.X.); 2The Connecticut Agricultural Experiment Station Valley Laboratory, Windsor, CT 06095, USA; Dewei.Li@ct.gov

**Keywords:** rhizosphere beneficial fungi, *Pinus massoniana*, genealogical concordance phylogenetic species recognition, one new taxon (*Talaromyces nanjingensis* sp. nov)

## Abstract

Rhizosphere fungi have the beneficial functions of promoting plant growth and protecting plants from pests and pathogens. In our preliminary study, rhizosphere fungus JP-NJ4 was obtained from the soil rhizosphere of *Pinus massoniana* and selected for further analyses to confirm its functions of phosphate solubilization and plant growth promotion. In order to comprehensively investigate the function of this strain, it is necessary to ascertain its taxonomic position. With the help of genealogical concordance phylogenetic species recognition (GCPSR) using five genes/regions (ITS, *BenA*, *CaM*, *RPB1*, and *RPB2*) as well as macro-morphological and micro-morphological characters, we accurately determined the classification status of strain JP-NJ4. The concatenated phylogenies of five (or four) gene regions and single gene phylogenetic trees (ITS, *BenA*, *CaM*, *RPB1*, and *RPB2* genes) all show that strain JP-NJ4 clustered together with *Talaromyces brevis* and *Talaromyces liani*, but differ markedly in the genetic distance (in *BenA* gene) from type strain and multiple collections of *T*. *brevis* and *T*. *liani*. The morphology of JP-NJ4 largely matches the characteristics of genes *Talaromyces*, and the rich and specific morphological information provided by its colonies was different from that of *T. brevis* and *T. liani*. In addition, strain JP-NJ4 could produce reduced conidiophores consisting of solitary phialides. From molecular and phenotypic data, strain JP-NJ4 was identified as a putative novel *Talaromyces* fungal species, designated *T. nanjingensis*.

## 1. Introduction

Rhizosphere fungi play roles in promoting plant growth and protecting plants from pests and pathogens. Phosphate-solubilizing fungi (PSF) are an important group of such fungi. Phosphate-solubilizing microbes in soil include PSF [1] and phosphate-solubilizing bacteria (PSB) [2]. Fungi and bacteria have their own advantages in adaptation in different environments. The variety and quantity of PSB were more than that of PSF [3], and the research studies on them are still in progress. Common PSF include *Aspergillus*, *Penicillium*, *Trichoderma*, and some mycorrhizal fungi. Phosphate-solubilizing fungi can be applied to a variety of crop ecosystems. For example, *Aspergillus niger* and *Penicillium chrysogenum* promote the growth and nutrient uptake of groundnut (*Arachis hypogaea*) [4]. Inoculation with the PSF *Aspergillus niger* significantly increases growth, root nodulation, and yield of soybean plants [5]. Phosphate-solubilizing fungi can also be applied to forest ecosystems. The fungal suspension and extracellular metabolites of *Penicillium guanacastense* have shown to increase the shoot length and root crown diameter of *Pinus massoniana* seedlings [6].

*Penicillium* is one of the most common genera of fungi worldwide. It is widely distributed in nature and primarily functions in breaking down organic matter to provide nutrients for its growth [7,8]. Since Link (1809) introduced the species concept of *Penicillium* [9] and Dierckx [10] introduced the subgenus classification system of *Penicillium*, studies on *Penicillium* have become increasingly popular. At the beginning of the 20th century, an early system of classification and identification based on colony characteristics and conidiophore branching patterns was proposed. The genus *Talaromyces* was first introduced by Benjamin (1955) as the sexual state of the genus *Penicillium* [11]. Stolk and Samson (1972) divided *Talaromyces* into four sections based on differences in their asexual states [12]. Later, more advanced and novel classification schemes based on conidiophore structure, branching pattern, and phialide shape, as well as strain growth characteristics, emerged. Pitt (1979) classified *Penicillium* into four subgenera: *Aspergilloides*, *Biverticillium*, *Furcatum*, and *Penicillium*, which contain 10 sections and 21 series. Since then, the modern concept of *Penicillium sensu lato* has emerged [13].

With the popularization of DNA-based phylogenetic studies of fungi, it has been gradually recognized that the subgenus *Biverticillium* within the genus *Penicillium sensu lato* is phylogenetically separate from other subgenera of *Penicillium* and is closely related to *Talaromyces*, the previously mentioned sexual morph of *Penicillium*. The subgenera *Aspergilloides*, *Furcatum*, and *Penicillium* originated from *Penicillium*
*sensu lato*, together with the genus *Eupenicillium*, and other species now fall within *Penicillium sensu stricto*, whereas subgenus *Biverticillium* is synonymized under the current genus *Talaromyces* [14,15]. Today, section *Talaromyces* is not limited to sexual species, but it still contains most of the sexually reproducing species in the genus *Talaromyces*. Yilmaz et al. (2014) proposed a new sectional classification for the genus *Talaromyces*, placing the 88 accepted species into seven sections, namely, *Bacillispori, Helici, Islandici, Purpurei, Subinflati, Talaromyces*, and *Trachyspermi* [15]. *Talaromyces flavus* (Klöcker) Stolk and Samson (= *T. vermiculatus* (P.A. Dang.) C.R. Benj.) has always been the typus of genus in the *Talaromyces* and *Talaromyces* section *Talaromyces* through many revisions of the genus *Talaromyces* [12,13,15]. The current latest concept of species in *Talaromyces* section *Talaromyces* is consistent with what Stolk and Samson (1972) described. Stolk and Samson (1972) introduced the *Talaromyces* section to include species that produce yellow ascomata, which can occasionally be white, creamish, pinkish, or reddish and yellow ascospores. Conidiophores are usually biverticillate-symmetrical, with some species having reduced conidiophores with solitary phialides. Phialides are usually acerose, with a small proportion of species having wider bases [12]. Section *Talaromyces* species are commonly isolated from soil, indoor environments, humans with talaromycosis and food products. Common species include *T. flavus*, *T. funiculosus*, *T. macrosporus*, *T. marneffei*, *T. pinophilus*, and *T. purpurogenus*.

Micromorphological features such as asexual sporulation structures (e.g., conidiophore) and sexual sporulation structures (e.g., cleistothecium) were of great significance for taxonomy. The branching pattern of conidiophores, namely the type of penicillus, is an important reference index for the traditional classification methods of *Penicillium* and *Talaromyces* fungi. The branching pattern generally includes Monoverticillate, Biverticillate, Terverticillate, Quaterverticillate, and Conidiophores with solitary phialides and Divaricate [13,16,17,18]. Although the classification of *Penicillium* (and *Talaromyces*) based on these branching patterns is not completely consistent with the classification status of *Penicillium* (and *Talaromyces*) in modern taxonomy, an accurate description of these morphological and structural characteristics is still considered important. The important micromorphology characteristics of *Penicillium* and *Talaromyces* fungi include the following: all components of conidiophore (stipes, ramus, ramulus, metula, and phialide) and the sizes, wall texture/ornamentation, color of conidium, ascocarp, ascus, ascospore, and sclerotium. The penicillus includes four parts: ramus, ramulus, metula, and phialide. Sclerotium is produced only under certain conditions; if there are any, observe and record it.

A breakthrough period in the rapid development of classification systems came with the advent of DNA sequencing technology in the 1990s. The identification of *Penicillium*-group and filamentous fungi began to shift from observation of morphological characteristics to molecular phylogeny. Morphological features are the physical structures with which an organism operates and adapts to its environment, and some features may differ or may be affected by specific factors in the surrounding environment. The effects of medium preparation, inoculation techniques, and culture conditions can be minimized by using strictly standardized protocols [19,20,21]. Morphological identification still plays an irreplaceable role in the fine identification of strains, and a polyphasic approach using both techniques was finally adopted. 

In *Penicillium*, *Talaromyces*, and many other genera of ascomycetes, internal transcribed spacer (ITS) sequences have been used to classify strains into species complexes or sections, as well as for species identification [15,18]. Due to the limitations of species barcoding based on the ITS region, secondary barcodes or identification markers are often required to identify isolated strains to the species level. Secondary barcodes should be easily amplified, able to distinguish closely related species, and come with a complete reference dataset (including representative gene sequences of all species). The following barcodes can generally be used for the identification of *Talaromyces* species. The Internal Transcribed Spacer (ITS) rDNA sequence is accepted as the official barcode for fungi [22]. β-tubulin (*BenA*) is used for the accurately identification of *Penicillium* species and can also be applied to *Talaromyces* species [15,18]. Trees have been constructed using other DNA barcode markers (Calmodulin (*CaM*), DNA-dependent RNA polymerase II (beta) largest subunit (*RPB1*), and DNA-dependent RNA polymerase II (beta) second largest subunit (*RPB2*)). Among these, *CaM*, *RPB1*, and *RPB2* exhibit the same potential as *BenA* and can be used as secondary barcodes for species identification. In recent years, usage of the *CaM* gene has gradually increased, and its reference dataset has become relatively complete. *RPB1* and *RPB2* have the added advantage of lacking introns in the amplicon, allowing for robust and easy alignment when used for phylogenetic analysis, but they may be difficult to amplify. At present, the reference dataset for the *RPB2* gene of *Talaromyces* species is fairly robust, whereas that for the *RPB1* gene is still being improved. During phylogenetic tree construction, in addition to the reference sequences of ex-types, other multiple collections from the same species should be considered to cover possible sequence variations. Comparing ITS, *BenA*, *CaM*, *RPB1*, and *RPB2* sequences from a suspected new species with sequences of the same markers in related species can help to determine whether a species is new via genealogical concordance phylogenetic species recognition (GCPSR) [23]. This approach, which involved multigene phylogeny, morphological descriptions using macro-morphological and micro-morphological characters and analysis of extrolites, has been used to develop the polyphasic species concept of filamentous fungi such as *Penicillium* and *Talaromyces*. 

In our preliminary study, rhizosphere fungus JP-NJ4 was obtained from Masson pine rhizosphere soil and screened for phosphate solubilization and plant growth promotion [24]. Fungus JP-NJ4 has the potential to be used as an ecofriendly soil amendment for forestry and farming. With the aid of internal transcribed spacer (ITS) sequences, this strain was preliminarily identified as *Penicillium pinophilum* (which is now classified in the genus *Talaromyces* and has been renamed *Talaromyces pinophilus*). However, the variability of ITS sequences is insufficient to distinguish among closely related species [22]. To comprehensively investigate the function of fungus JP-NJ4, the classification status of this strain was investigated further. The identification process for strain JP-NJ4 involved many standard strains (type strains) that are currently stored at the Central Bureau of Fungal Cultures (Centraalbureau Voor Schimmelcultures (CBS)), which is part of the Royal Netherlands Academy of Arts and Sciences and was founded in 1904 by the Association Internationale des Botanistes [25]. Currently, CBS is one of the largest mycological research centers in the world, with more than 60,000 species in cultivation, including the type strains of many filamentous fungus and yeast species. Here, after reviewing the literature and observing the characteristics of fungus JP-NJ4, this strain was identified and described by referring to the standard research method (GCPSR) recommended in previous international research on filamentous fungal species such as *Penicillium* and *Talaromyces*, etc. 

## 2. Materials and Methods

### 2.1. Source of the Strain

The strain JP-NJ4 was a phosphate-solubilizing fungus isolated from rhizosphere soil of *Pinus massoniana* (yellow brown soil) in the back mountain of Nanjing Forestry University. The strain is now stored in the China Center for Type Culture Collection (CCTCC) (http://www.cctcc.org, accessed on 18 January 2022). Holotype with the preservation number M 2012167 was stored in a metabolically inactive state by cryopreservation [26,27].

### 2.2. DNA Extraction, PCR Amplification, and Sequencing of Strain JP-NJ4

Strain JP-NJ4 was cultured on malt extract agar (MEA) culture medium at 25 °C for 7–14 days. Genomic DNA was extracted and purified according to the method of Cubero et al. [28], and the extract was stored at −20 °C. The DNA barcode markers required for the identification of JP-NJ4 strain included the ITS region and *BenA*, *CaM*, *RPB1*, and *RPB2* genes [29,30,31,32,33,34,35,36,37,38]. The primers needed for the amplification of these genes are shown in Appendix A. All primers and polymerase chain reaction (PCR) amplification sequences needed for the experiment were synthesized and sequenced by the Shanghai Sangon Company (http://www.sangon.com, accessed on 18 January 2022). 

In this study, a 50.0 μL DNA amplification thermal cycling reaction mixture system was selected, and the formula of 20.0 μL reaction system was also provided. The volumes of the components in the system are as follows: premix Taq™ solution 25.0 μL, DNA template (10 ng/μL) 2.5 μL, forward primer 2.5 μL, reverse primer 2.5 μL and dd H_2_O 17.5 μL for 50.0 μL system; premix Taq™ solution 10.0 μL, DNA template (10 ng/μL) 1.5 μL, forward primer 1.0 μL, reverse primer 1.0 μL, and dd H_2_O 6.5 μL for 20.0 μL system. Premix Taq™ (Ex Taq ™ Version 2.0 plus dye) is a 2x concentration mixed reagent of DNA polymerase, buffer mixture, and dNTP mixture required for PCR reactions purchased from Takara company (https://takara.company.lookchem.cn/, accessed on 18 January 2022). The concentration of the ingredients in the Premix Taq™ solution is as follows: Ex Taq Buffer (2×conc.) with Mg^2+^ at a concentration of 4mM (mmol/L); highly efficient amplification DNA polymerase (TaKaRa Ex Taq) at a concentration of 1.25 U/25 μL; the dNTP (deoxy-ribonucleoside triphosphate) Mixture (2×conc.), with a concentration of 0.4 mM (mmol/L) for each base; additional pigment markers (Tartrazine/Xylene Cyanol FF), specific gravity additaments, and stabilizers were included. The reagent is stored at −20 °C. The total amount of DNA template can be 10–100 ng, and it can be added according to the experimental requirements. The concentration of primer prepared in accordance with the operational guidelines is 100 μmol/L, diluted 10 times to 10 μmol/L for use.

The DNA amplification thermal cycling programs for each gene is as follows: Standard PCR was selected for general ITS, *BenA*, and *CaM*, with initial denaturing 94 °C for 5 min, cycles 35 of denaturation 94 °C for 45 s, annealing 55 °C (52 °C) for 45 s, elongation 72 °C for 60 s, final elongation 72 °C for 7 min, and rest period 10 °C, ∞. Touch-down PCR was selected for *RPB1*, with 5 cycles of 30 s denaturation at 94 °C, followed by primer annealing for 30 s at 51 °C, and elongation for 1 min at 72 °C; followed by 5 cycles with annealing for 30 s at 49 °C and 30 cycles for 30 s at 47 °C, finalized with an elongation for final 10 min at 72 °C, rest period 10 °C, ∞ (the denaturation and elongation conditions of the second and third cycles are the same as those of the first cycle). Touch-up PCR (= step-up PCR) was selected for *RPB2*, with initial denaturing 94 °C for 5 min, followed by 5 cycles of 45 s denaturation at 94 °C, primer annealing for 45 s at 50 °C (48 °C), and elongation for 1 min at 72 °C; followed by 5 cycles with annealing for 45 s at 52 °C (50 °C) and 30 cycles for 45 s at 55 °C (52 °C), finalized with an elongation for final 7 min at 72 °C, rest period 10 °C, ∞ (the denaturation and elongation conditions of the second and third cycles are the same as those of the first cycle). The values in parentheses refer to alternative reaction conditions.

### 2.3. Phylogenetic Tree Construction of Strain JP-NJ4

Sequences of five genes from strain JP-NJ4 have been sequenced and deposited in GenBank (Table 1). By conducting a Basic Local Alignment Search Tool (BLAST) search in National Center for Biotechnology Information (NCBI) database (https://www.ncbi.nlm.nih.gov, accessed on 18 January 2022), the results showed the best matched DNA sequences for each gene/region. In order to make phylogenetic trees, the type strains of *Talaromyces* species were added. For monogenic and polygenic phylogeny, ITS, *BenA*, *CaM*, *RPB1*, and *RPB2* sequence data were compared and aligned using ClustalW software included in the MEGA package version 6.0.6 [39]. All datasets (DNA sequences) were concatenated in MEGA and the BioEdit Sequence Alignment Editor software (Version 7.0.9.0) [40]. The aligned data sets were analysed using both Maximum Likelihood (ML) and Bayesian inference (BI) methods, and ML phylogenetic trees were constructed for each gene/region and concatenated polygenic sequences. According to the results of Akaike Information Criterion (AIC) calculated in MEGA package, the best model for ML phylogenetic tree construction is selected. The ML analysis is performed, and the trees were constructed by calculating the initial tree (constructed by the BioNJ method), selecting the Nearest-Neighbour-Interchange (NNI) option for the following heuristic search. Bootstrap analysis was performed on 1000 repetitions to calculate the support at the node. Bayesian Inference phylogenies were inferred using PhyloSuite v1.2.1 [41]. ModelFinder was used to select the best-fit model (2 parallel runs, 2,000,000 generations) using Bayesian Information Criterion (BIC) for BI [42]. The sample frequency was set at 100, with 25% of trees removed as burn-in. Bayesian inference posterior probabilities (BIpp) values and bootstrap values are labelled on nodes. 

Table 1 summarised the information of type strains and other related strains, including collection numbers, source and location of strains, and GenBank accession numbers of five genes/regions (ITS barcode and four auxiliary molecular markers: *BenA*, *CaM*, *RPB1*, and *RPB2*) used for phylogenetic analysis of strain JP-NJ4. According to the information of Samson et al. (2011) and Yilmaz et al. (2014) [14,15], the type strains and other related strains were selected. The current sectional classification information of *Talaromyces* species was also marked in Table 1.

### 2.4. Observation on the Morphological Characteristics of Strain JP-NJ4

Important features used to describe the large group of *Penicillium* and its related fungi are as follows: Macromorphology, including colony texture, mycelium growth and color, shape, color, abundance, and texture of conidia, the presence and color of soluble pigments and exudates, the reverse color of the colony and the acid production of the strain on creatine sucrose agar (CREA) [43], etc. Micromorphology, including asexual sporulation structures (e.g., conidiophore) and sexual sporulation structures (e.g., cleistothecium), etc. To comprehensively investigate the growth of strain JP-NJ4 on different media, we formulated the following supplemented-medium types (see Appendix A) from common media, these media can be used to observe other taxonomic characteristics of strains. 

Czapek yeast autolysate (CYA) [13] and malt extract agar (MEA) [8] are two standard media recommended for species identification of *Penicillium* and related filamentous fungi. Czapek’s agar (CZ) [16] CZ is the medium used by Raper and Thom (1949) and Ramírez (1982) in taxonomic studies [44]; this also includes Blakeslee’s Malt extract agar (MEAbl) of Blakeslee (1915) [45]; yeast extract sucrose agar (YES) [43]; and YES as the recommended medium for the analysis of species’ extracellular secretions (extrolites). Oatmeal agar (OA) [8] and Hay infusion agar (HAY) medium [46] were also included. Sexual reproduction of fungal strains most often occurs on OA and HAY media, which can provide valuable information for taxonomy. Use oatmeal/flakes for OA and dry straw for HAY. Creatine sucrose agar (CREA) is the production of acid that can be observed by color reactions (ranging from purple to yellow) in CREA, which are often useful for distinguishing closely related species. Dichloran 18% Glycerol agar (DG18) [47] and Czapek Yeast Autolysate agar with 5% NaCl (CYAS) [18] were used. DG18 and CYAS were used to detect the growth rate of the strain under low water activity.

Preparation for macromorphology observation: The strain JP-NJ4 was inoculated in Potato dextrose agar (PDA) medium and cultured at 15 °C for 25 days to collect conidia. Conidia were washed with distilled deionized water (dd H_2_O) and diluted with a semi-solid agar solution containing 0.2% agar and 0.05% Tween 80 to prepare the conidia suspension, which was stored at 4 °C for standby use [13]. Conidia suspension was extracted with a micropipette (Eppendorf) and inoculated in three points (1 μL per point) [18]. All media were incubated at a constant temperature of 25 °C for 7 days; each formula of medium is shown in Appendix A. In addition, the Czapek Yeast Autolysate agar (CYA) was cultured at 30 °C and 37 °C, and Malt Extract agar (MEA) was cultured at 30 °C, and the data were recorded for the species identification of strain JP-NJ4. After 7 and 14 days of strain culture, the criss-cross method was used to measure the colony diameter.

Preparation for the micromorphology observation: Colonies of strain JP-NJ4 cultured on MEA for one to two weeks in a dark environment at 25 °C were used for micromorphology observation, and OA and HAY medium were used when ascomata were not observed on MEA. OA and HAY media are often used for the observation of ascocarp, ascus and ascospore [31,48,49,50] and may be cultured for up to 3 weeks if required for ascocarp production. Then, the colonies used for micromorphology observation were rinsed with 2mL 0.1 mol/L phosphate-buffered saline (PBS) three times. Lactic acid (60%) was used as the fixative. Since most species produce large amounts of hydrophobic conidia, 70% ethanol is usually used to flush out excess conidia and prevent air from getting trapped in lactic acid between the slide and cover. The characteristics of strain JP-NJ4 were observed with a compound microscope (Axio Imager M2.0; Zeiss, Germany) equipped with a digital camera (AxioCam HRc; Zeiss, Germany). The colonies were dehydrated in a graded ethanol solution and dried with liquid carbon dioxide at a critical point (EmiTech K850). After gold spraying (Hitachi E-1010), the Micromorphology of strain JP-NJ4 (conidiophore, conidium, ascocarp, ascus, and ascospore) was observed by scanning electron microscope (SEM) (FEI Quanta 200, FEI, USA). 

## 3. Results

### 3.1. Taxonomy of Strain JP-NJ4

From the molecular and phenotypic data, it can be inferred that the strain JP-NJ4 belongs to *Talaromyces*. We identified it as a putative new species (new taxon) here [27,51]. 

#### Taxonomy

***Talaromyces nanjingensis*** X.R. Sun, X.Q. Wu and W. Wei, sp. nov. (this study).

MycoBank (No: MB837590).

**Etymology**: Latin, ‘*nanjingensis*’ refers to Nan jing, the name of the city where the species originated.

**Typus (Type strain)**: China, Jiangsu, Nanjing, on the rhizosphere soil from *Pinus massoniana*, 11 April 2011, W. Wei, deposited in China Center for Type Culture Collection (CCTCC) (Collection number. CCTCC M 2012167) (http://www.cctcc.org, accessed on 18 January 2022). Holotype: CCTCC M 2012167. Culture ex-holotype: CCTCC M 2012167.

**Distribution**: Area of Nanjing, China.

**Habitat**: Rhizosphere soil from *Pinus massoniana*.

**ITS barcode:** MW130720. (Alternative markers for identification: *BenA* = MW147759; *CaM* = MW147760; *RPB1* = MW147761; *RPB2* = MW147762).

In: Talaromyces section Talaromyces

**Colony diam, 7 d (mm)**: CZ 29–33; CYA 25 °C 25–29; CYA 30 °C 30–37; CYA 37 °C 21–31; MEA 25 °C 31–33; MEA 30 °C 35–41; MEAbl (34–43); OA 38–44; DG18 15–18; CYAS No growth; YES 30–40; CREA 18–24; HAY No growth. 

**Colony characters**: The top and reverse colony morphology of the strain *Talaromyces nanjingensis* in different media was described. CYA 25 °C, 7 d: top colonies raised at the centre, yellow and margins white; margins low, plane, entire; texture velvety to floccose; sporulation absent to sparse; a small amount of yellow and orange soluble pigments present at 25, 30 and 37 °C; exudates absent; reverse centre pastel yellow (2D4) to pale yellow (1A4); 25 °C, 14 d: top colonies centre pale yellow (1A4) and margins white; the amount of orange exudates present on colonies centre; 30 °C, 14 d: top colonies white, pastel yellow and pinkish-red; a small amount of orange exudates present on colonies centre; formation of yellow ascomata; 37 °C, 14 d: top colonies greyish green (28C5); orange exudates abundant on the gully formed by colony bulge. MEA 25 °C, 7 d: top colonies low, plane; margins low, plane, entire (2–3 mm); white and yellow; texture velvety to floccose; sporulation sparse to moderately dense, conidia greyish-green (26B4-26C4); weak yellow and orange soluble pigments present; exudates absent; reverse light orange to light yellow (5A5-4A5); 25 °C, 14 d: top colonies centre pale yellow (1A4) and margins greyish-yellow (1B4); reverse dark brown (9F8) centre fading into reddish-brown (9E8) to greyish orange (5C5) at margins; a large amount of red soluble pigments present; 30 °C, 14 d: formation of yellow ascomata. YES 25 °C, 7 d: top colonies raised at the centre, sulcate; margins low, plane, entire (3–4 mm); light yellow (4A5) and margins white; texture velvety to floccose; sporulation absent, conidia dull green to greyish green (25D4-25D5); soluble pigments absent; exudates absent; reverse centre pastel yellow and margins white; 25 °C, 14 d: top colonies centre white and margins light yellow (4A5); reverse deep yellow and deep orange (4A8-5A8) to light yellow (2A5); a small amount of yellow soluble pigments present. DG18 25 °C, 7 d: top colonies slightly raised at the centre, plane; margins low, plane, entire (2 mm); pastel green (28A4) and margins white; texture floccose; sporulation moderately dense, conidia greyish green to dull green (25D5-25E4); soluble pigments absent; exudates absent; reverse centre dark green (28F5) and margins white; 25 °C, 14 d: top colonies centre greyish green (28C5) and margins white, reverse pale light green (1B3) to white. OA 25 °C, 7 d: top colonies raised at the centre, plane, formation of yellow ascomata (abundant at 25 °C, 14 d); margins low, plane, entire (2–3 mm); white and yellow; sporulation absent; soluble pigments absent; exudates absent; reverse pastel yellow (2D4). CREA 25 °C, 7 d: acid production present strong; 25 °C, 14 d: acid production present very strong; mycelia all weak at 7 d and 14 d.

**Micromorphology**: Conidiophores monoverticillate and biverticillate; it also produces reduced conidiophores consisting of solitary phialides. Stipes smooth-walled, 20–100 × 2.5–3 μm; branches 8–20 μm; metulae two to five, divergent, 7–16 × 2.5–3 μm; phialides acerose, two to five per metulae, 6–8 × 2–3 μm; Conidia smooth, globose to subglobose, 2–3 × 3 μm, sometimes ovoid, 3 × 3–3.5 μm. Ascomata mature after one week of incubation on OA, two weeks of incubation on CZ at 25 °C and on CYA and MEA at 30 °C. Ascomata yellow, globose to subglobose, 300–950 × 300–1000 μm, Asci, which are irregular in shape and size depending on the number of ascospores inside them, 10–12 × 8–10 μm; Ascospores, the shape and size are uniform and stable, broadly ellipsoidal, spiny, 3.5–5 × 2–3 μm.

**Distinguishing characters**: *Talaromyces nanjingensis* produces relatively fast-growing colonies (Colony diam (mm)) on MEA (31–33), CYA (25–29) and YES (30–40) at 25 °C (faster at 30 °C, MEA 35–41, CYA 30–37), as well as the fastest-growing colonies on MEAbl (34–43) and OA (38–44) at 25 °C. It produces yellow ascomata on CZ and OA media with spiny ellipsoidal ascospores, similar to those of *T. austrocalifornicus*, *T.*
*flavovirens*, *T.*
*flavus*, *T. macrosporus*, *T. muroii*, *T. thailandensis*, and *T. tratensis*. On colony size at 25 °C on CYA and MEA after 7 d (CYA 25–29; MEA 31–33), *T. nanjingensis* is more similar to *T. aculeatus*, *T. angelicus*, *T. dendriticus*, *T. indigoticus*, *T. panamensis*, *T. varians*, and *T. siamensis*. According to the phylogenetic tree, *T. nanjingensis* and *T. liani* are clustered together. *T. nanjingensis* produces yellow ascomata, whereas *T. brevis* and *T. liani* produce yellow to orange and yellow to orange-red ascomata on OA medium, respectively. *Talaromyces nanjingensis*, *T. brevis* and *T. liani* both have ellipsoidal ascospores. *Talaromyces nanjingensis* grows more faster and produces more acid on CREA than *T. brevis* and *T. liani*.

### 3.2. Phylogeny-Based Species Identification

With the help of concatenated phylogenetic trees based on five gene regions, including the internal transcribed spacer region, *BenA*, *CaM*, *RPB1*, and *RPB2*, we investigated the taxonomic position of strain JP-NJ4. Figure 1, Figure 2 and Figure 3 and Appendix A show the phylogenetic relationships among strain JP-NJ4 and representative species of *Talaromyces.* Concatenated phylogenetic trees of five (ITS, *BenA*, *CaM*, *RPB1*, and *RPB2*) and four (ITS, *BenA*, *CaM*, and *RPB2*) gene regions and individual phylogenetic trees of each gene region were constructed using the maximum-likelihood method. *Talaromyces dendriticus* (CBS_660.80_T) was chosen as an out-group for *Talaromyces* section *Talaromyces*. *Trichocoma paradoxa* (CBS_788.83_T) was chosen as the out-group for the *Talaromyces* genus. Bootstrap values obtained from 1000 replications are shown at the nodes of the tree, and bootstrap support lower than 50 is not shown. In the multi-gene phylogenetic analysis (five gene region), strain JP-NJ4 clustered with *T. liani* (Figure 1) in *Talaromyces* section *Talaromyces* (orange area), with bootstrap values of 100% (BIpp = 1). The concatenated phylogeny of five gene region shows that strain JP-NJ4 and *T. liani* differ in their genetic distance from other species of *Talaromyces*. Phylogenetically, the results of four genes indicate that strain JP-NJ4 *T. nanjingensis* is close to *T. brevis*, with bootstrap values of 93% (BIpp = 1) (Figure 2).

Among the phylogenetic trees obtained from each DNA gene region, that of the ITS region was less clearly resolved; although most species formed monophyletic groups in the strict consensus trees, several had low bootstrap support values. The ITS sequence of strain JP-NJ4 clustered with those of nine other strains of *T. liani* and three strains of *T. brevis* (bootstrap = 32%, BIpp = 0.61) (Appendix A). The *CaM* sequence of strain JP-NJ4 clustered well with two strains of *T. brevis* (bootstrap = 65%, BIpp = 0.99) (one strain of *T. brevis* was deleted because its sequence was shorter) and seven other strains of *T. liani* (bootstrap = 99%, BIpp = 0.96) (Appendix A). The *RPB1* sequence of strain JP-NJ4 clustered with that of the type strain of *T. liani* (CBS_225.66_T) (bootstrap = 85%, BIpp = 0.99) (Appendix A). The *RPB2* sequence of strain JP-NJ4 clustered perfectly with three strains of *T. brevis* (bootstrap = 99%, BIpp = 1) (Appendix A). The phylogenetic tree of the *CaM* gene region shows that strain JP-NJ4 and *T. liani* differed little in their genetic distance from other species of *Talaromyces*. However, the phylogenetic trees of the *BenA*, *RPB1*, and *RPB2* gene regions show that strain JP-NJ4 and *T. liani* differed markedly in their genetic distance from type strain of *T. liani* and other multiple collections of *T. liani*.

The result of single gene *BenA* indicate that *T. nanjingensis* and ‘*T. liani*’ (voucher KUC21412) are clustered together (Bootstrap 88%/ BIpp 1), and both of them have lower bootstrap values (Bootstrap 45%/ BIpp -) with *T. liani* (CBS_118885) and higher bootstrap values (Bootstrap 63%/BIpp 0.99) with nine other strains of *T. liani* at the node (Figure 3). This indicates that *T. nanjingensis* is still genetically different from its genetic relatives *T. liani* and *T. brevis*. The sequences of *T. nanjingensis* and ‘*T. liani*’ (voucher KUC21412) were significantly similar in *BenA* gene. However, *T. nanjingensis* and ‘*T. liani*’ (voucher KUC21412) differ from *T. liani* and *T. brevis* in *BenA* gene by more than ten bases, and half of their base arrangement pattern is similar to *T. liani* and the other half is similar to *T. brevis* (Appendix A). This could mean they should be new species. It also explains the low bootstrap values. This is also due to the continued discovery of new species, filling gaps in the evolutionary trees, and the lack of some transitional species, of which the *T. nanjingensis* is one, which has characteristics common to both *T. liani* and *T. brevis*. *T. nanjingensis* is more similar to *T. brevis* in acid production. *T. liani* (CBS_118885) is the only acid-producing strain of *T. liani* that is genetically closest to *T. nanjingensis*. The phenotypic information of these species may also hint at evolutionary continuity. After a detailed search, we found that *T.*
*liani* strain T2C1 is equivalent to ‘*T. liani*’ (voucher KUC21412) and ITS sequence was also obtained. ‘*T. liani*’ (voucher KUC21412) has at least two base differences with *T. nanjingensis*, *T. liani*, and *T. brevis* in the ITS region, and the front-end of the sequence is similar to that of *T. liani* and *T. brevis* (CBS_141833_T). In particular, it has a distinctive differential base A at the end of its sequence (Appendix A). Therefore, ‘*T. liani*’ (voucher KUC21412) is also different from *T. nanjingensis*. ‘*T. liani*’ (voucher KUC21412) is described by Heo et al. as one of the microorganisms selected from intertidal mudflats and abandoned solar salterns that can produce bioactive compounds [52]. The strain voucher KUC21412 was described as ‘*T. liani*’ (with quotation marks) in this manuscript, as its current species identity may be in some doubt. ‘*T. liani*’ (voucher KUC21412) is a comparable species, although only ITS (MN518409.1) and *BenA* sequences (MN531288.1) have been submitted to NCBI, the quality of the sequences is reliable, which proves that the *BenA* sequence of *T. nanjingensis* is reliable. Although *T. nanjingensis* had little difference with *T. brevis* in ITS, *CaM*, *RPB1*, and *RPB2* genes, it had great difference with *T. brevis* in *BenA* gene (Appendix A). *BenA* is the secondary barcode with the highest reliability in filamentous fungi; thus, the classification results obtained by using this gene are relatively more referential and accurate [15,18]. According to the Bayesian tree (ITS + *BenA* + *CaM* + *RPB2* and single gene *BenA*) and the ML tree with deletion of *T. brevis* and ‘*T. liani*’ (voucher KUC21412), it can be more obvious that *T. nanjingensis* has a long genetic distance from *T. brevis* and *T. liani* (the small diagrams in Figure 2 and Figure 3).

More obviously, the phylogenetic tree of *BenA* gene with long sequence version (Figure 3B) was better than that of the *BenA* gene with short sequence version (Figure 3A) to show the true classification status of strain JPNJ4. The results showed that strain JPNJ4 was quite different from *T. brevis* and *T. liani* in phylogeny (Figure 3B). The long sequence version of the phylogenetic tree (Figure 3B) only reduced or lost information on some other species, but this did not affect the accurate identification of JP-NJ4, because this version included *T. brevis* and *T. liani*. New species resources are undoubtedly important, as are innovations in identification methods and fine-delineation of species. Taxonomy of these species of *Talaromyces* are similar to the results obtained by Samson et al. (2011) and Yilmaz et al. (2014) [14,15]. These phylogenetic results suggest that strain JP-NJ4 is a potential novel species.

### 3.3. Species Identification Based on Macromorphology and Micromorphology

By combining these results with morphological observations, the taxonomic position of strain JP-NJ4 can be further elucidated. The macromorphology of strains, including the morphology and diameter of colonies on specific media, is an important trait for species identification. Based on the preliminary results on CYA and MEA, the macromorphology of the strain may be observed on several other culture media for more accurate identification.

We selected as many media as possible to observe the strains in more detail. Czapek (CZ) medium was used in early taxonomic studies of *Penicillium* and was selected for comparison with CYA medium. Blakeslee’s MEA, which has been widely used historically, was compared with the MEA culture medium used by the CBS-KNAW Fungal Biodiversity Centre in Utrecht (Appendix A). Medium with the addition of hay (HAY) was compared with oatmeal agar for observing the sexual reproduction of fungal strain JP-NJ4. However, strain JP-NJ4 did not grow on this recommended medium.

To clearly observe the colony morphology of strain JP-NJ4 on various culture media, we obtained photographs with black and white background colors. In this paper, we provided colony morphology photographs of strain JP-NJ4 grown at 25 °C for 7 (Figure 4 and Appendix A) and 14 d (Figure 5 and Appendix A) on 10 different media, as well as image data for strain JP-NJ4 grown at 25 °C, 30 °C, and 37 °C in CYA medium (Appendix A).

Reverse colonies of *Talaromyces* species on CYA and MEA media commonly produce yellow or red soluble pigments. Numerous species of *Talaromyces*, including *T. albobiverticillius*, *T. amestolkiae*, *T. atroroseus*, *T. cnidii*, *T. coalescens*, *T. marneffei*, *T. minioluteus*, *T. pittii*, *T. purpurogenus*, *T. ruber*, and *T. stollii* produce soluble red pigments. In *T. nanjingensis* strain JP-NJ4, weak production of yellow and orange soluble pigments was observed on CYA and MEA in colonies grown at 25 °C for 7 d (Figure 4 and Appendix A). The reverse colony color on MEA was similar to those of *T. albobiverticillius*, *T. minioluteus*, and *T. purpurogenus*. Strong and stable red soluble pigment production occurred on MEA in colonies grown at 25 °C for 14 d (Figure 5 and Appendix A). The reverse colony color on MEA in colonies grown at 25 °C for 14 d was similar to those of *T. amestolkiae*, *T. coalescens*, and *T. marneffei* grown on MEA at 25 °C for 7 d. *T. nanjingensis* strain JP-NJ4 could produce acid on CREA at 25 °C for 7 d (present strong, the color reaction showed a marked shift from purple to yellow) and at 25 °C for 14 d (present very strong; the color reaction appears to be intense yellow).

The macromorphologies of *T. nanjingensis* strain JP-NJ4, *T. liani*, and *T. brevis* on various media were obviously different in terms of colony growth rate and mycelia color. In terms of colony growth rate, the difference between the three species on MEA and CREA medium was the greatest. In an ascending order of growth rate, we have MEA medium (25 °C, 7 d), *T. nanjingensis* strain JP-NJ4 (31–33), *T. liani* (35–45), and *T. brevis* (50–51); in addition, we also have CREA medium (25 °C, 7 d), *T. liani* (10–20), *T. brevis* (13–14), and *T. nanjingensis* strain JP-NJ4 (18–24). In terms of mycelia color, on OA medium (25 °C, 7 d), the three species were sequentially *T. nanjingensis* strain JP-NJ4 (white and yellow), *T. brevis* (primrose), and *T. liani* (white and yellow) according to the color of mycelia from light to dark; on CYA medium (25 °C, 7 d), the order was *T. liani* (white and pastel yellow), *T. brevis* (white and flesh) and *T. nanjingensis* strain JP-NJ4 (yellow and margins white). Other detailed data are shown in Table 2.

*Talaromyces* species generally produce acerose phialides and ellipsoidal to fusiform conidia. *T. nanjingensis* strain JP-NJ4 produces reduced conidiophores consisting of solitary phialides (Figure 6A), most conidiophores are monoverticillate and biverticillate, and conidia are globose to subglobose and sometimes ovoid (Figure 6B–H). With the help of scanning electron microscopy, clearer pictures of conidia can be seen (Figure 6I,J). *Talaromyces liani* produces ellipsoidal conidia. *Talaromyces brevis* produces subglobose to fusiform conidia. In addition, some species of *Talaromyces* produce rough-walled, globose conidia, including *T. aculeatus*, *T. apiculatus*, and *T. verruculosus* (classified in sect. *Talaromyces*), as well as *T. diversus* and *T. solicola* (classified in sect. *Trachyspermi*).

Many species of *Talaromyces* have the ability to produce ascomata (ascoma = ascocarp; plural, ascomata) (Figure 7A–D). Generally, ascomata are yellow, but some species produce green (*T. derxii*, *T. euchlorocarpius*, and *T. viridis*) or creamish white ascomata (*T. assiutensis* and *T. trachyspermus*). The size, shape, and ornamentation of ascospores can be used to distinguish among species of *Talaromyces*. In most species of *Talaromyces*, ascospores are broadly ellipsoidal and spiny, but *T. bacillisporus* and *T. rotundus* have spiny globose ascospores and *T. tardifaciens* produces smooth globose ascospores. The ascospores of strain JP-NJ4 and *T. liani* are broadly ellipsoidal and spiny, and the ascospores of *T. brevis* are ellipsoidal and spiny. The ascospore sizes of *T. nanjingensis* strain JP-NJ4, *T. brevis*, and *T. liani* differed, at 3.5–5 × 2–3 μm (Figure 7E–I), 3.5–4.5 × 3–4, and 4–6 × 2.5–4 μm, respectively. The ascospores of *T. stipitatus* have single equatorial ridges, whereas those of *T. udagawae* have numerous ornamented ridges, and *T. helicus* has smooth ascospores.

According to the phylogenetic results, *T. nanjingensis* strain JP-NJ4 belongs to the genus *Talaromyces*. The taxonomic status of this strain can be further determined through description of its morphological characters. *Talaromyces liani* [15] and *Talaromyces brevis* [53] are the two species most closely related to *T. nanjingensis* strain JP-NJ4 in terms of molecular phylogeny, and they were selected as the control group for morphological comparison (Table 2). Table 2 contains summaries of the general macro-morphological and micro-morphological characters observed, including the most important characters: growth rates on different media, production of ascomata and soluble pigments, and acid production on creatine sucrose agar.

## 4. Discussion

*Talaromyces* species have a cosmopolitan distribution and have been isolated from a wide range of substrates. Soil is their main habitat, but new species have been obtained from indoor air, dust, clinical samples, plants, leaf litter, honey, and pollen [18,54,55,56,57,58,59,60,61,62]. *Talaromyces* species have positive impacts in the medical field. The members of this genus can produce a variety of antibiotics and antibacterial substances, such as the rugulosin produced by *T. rugulosus* [11,63]. Other extrolites of the genus (e.g., erythroskyrine, etc.) have anti-tumor [64], anti-malignant cell proliferation (antiproliferative), and anti-oxidant properties [65]. *Talaromyces* fungi also have a strong ability to produce enzymes, including that of β-rutinosidase and phosphatase [66,67], endoglucanase and cellulase [68], cellulase [69,70,71], and others. These fungi have also been investigated for functions in plant disease resistance, such as *T. flavus* [72,73,74,75] and *T. pinophilus* [76]; moreover, this includes the plant growth promotion of *T. pinophilus* [77]. In the present study, fungal strain JP-NJ4, which was isolated from the rhizosphere soil of *Pinus massoniana*, exhibited abilities of phosphate solubilization and plant growth promotion [24], and it was identified as a novel species in genus *Talaromyces,* section *Talaromyces*, using the polyphasic approach in this manuscript.

The fungal genera of *Penicillium* and *Talaromyces* have many similarities in morphology, such as asexual sporulation structures (e.g., conidiophore), the branching pattern of conidiophores, namely the type of penicillus, and sexual sporulation structures (e.g., cleistothecium). Mistakes are easily made when distinguishing between them. Therefore, we can use molecular methods to conduct preliminary identification of species in these genera. It should be noted, for the modern taxonomic identification of a species, morphological characteristics and molecular phylogenetic results are equally important. Professional recommendations regarding appropriate phylogenetic and morphological data in species delineation are necessary to avoid taxonomic discrepancies [78]. Phylogenetic trees of species are constructed using extensive data obtained through searches and literature review. Normally, the first step is to input a nucleotide sequence obtained through PCR and sequencing technology into the NCBI website for comparison using the nucleotide BLAST. Using the default settings, we obtained 100 sequences that are most similar to the target sequence. The purpose of this step is to roughly determine the genus of the unknown strain. In this paper, BLAST analysis was conducted using ITS, *BenA*, *CaM*, *RPB1*, and *RPB2* sequences of strain JP-NJ4.

During the process of collecting and collating the sequences needed to build phylogenetic trees, we encountered the following problems. Among the sequences submitted to NCBI, for the same gene from the same strain of the same species, sequences were uploaded under multiple sequence numbers. By conducting BLAST analysis of the sequence and preliminary phylogenetic tree construction, we found that some of the sequences were consistent with the earliest submitted sequences, whereas others did not cluster with the type strains of their species. This difference may be due to misidentification by later sequence submitters or mislabeling of different strains as the same strain. Therefore, when selecting sequences to construct phylogenetic trees, if two sequences are obtained with differing base compositions, we used the sequences submitted earlier or those referenced in the authoritative literature. Only using validated sequences is also reliable. If the sequences were identical, they were all retained in the tables used to build the phylogenetic tree (Table 1).

The ITS region is the most commonly used molecular marker for fungal identification. In *T. liani*, NRRL 1014 and NRRL 1015 are equivalent to NRRL 1009, and the base sequences of the ITS region and the other four specific genes in the three strains are identical. Therefore, when constructing the ITS phylogenetic tree for strain JP-NJ4, NRRL 1009 was selected to represent all three strains [79]. In addition, nine other *T. liani* strains were added. By conducting sequence alignment analysis of the *CaM* gene, we found notable differences in the composition and arrangement of the bases in this gene among species in different sections of genus *Talaromyces*. This may result in the deletion of too many bases in order to ensure sequence alignment in the tree constructing of strain JP-NJ4 at the genus level, resulting in loss of information. Therefore, in order to ensure the length of a *CaM* sequence in tree construction and improve the accuracy of species identification, the *CaM* gene phylogenetic tree was constructed at the level of section *Talaromyces*. We further determined the taxonomic status of strain JP-NJ4 by evaluating the taxonomic relationships among these highly similar species within the genus *Talaromyces*. In addition, during the Alignment-Align process of ClustalW in MEGA software (Version 6.0 and 7.0), inaccuracies may be introduced into the alignment results when large differences exist among the sequences. Therefore, the best comparison results can be obtained through multiple repeated comparisons.

We also found that the gene sequences of *RPB2* from some type strains of *Talaromyces* species could not be retrieved from the NCBI database. By performing comparison and analysis of the *RPB2* sequences of other species in genus *Talaromyces*, we found that the gene sequence data of RNA polymerase (RNA polymerase gene, partial cds) downloaded for these type strains included the gene sequence of *RPB2*. Therefore, these RNA polymerase gene sequences can be used to complement the construction phylogenetic trees based on the *RPB2* gene. Moreover, in previous studies of *Penicillium* and *Talaromyces* [14,15,18,31,80], the precedent of using the RNA polymerase gene sequence for constructing a *RPB2* phylogenetic tree has been established (e.g., JX315698 *Talaromyces amestolkiae* DTO 179F5_T). Using this method, the taxonomic status of unknown species can be further refined. The sequences used for this analysis include the following: KX961275 *Talaromyces angelicus* Korean Agricultural Culture Collection (KACC) 46611, KX961285 *T. aurantiacus* CBS 314.59, KX961283 *T. flavovirens* CBS 102801, KX961280 *T. galapagensis* CBS 751.74, KX961278 *T. indigoticus* CBS 100534, KX961282 *T. intermedius* CBS 152.65, KX961276 *T. muroii* CBS 756.96, KX961281 *T. oumae-annae* CBS 138208, JX315712 *T. stollii* CBS 408.93, and KX961279 *T. veerkampii* CBS 500.78. Some specific genes, such as Translation elongation factor (*Tef*) and mitochondrial Cytochrome c oxidase 1 (*Cox1*), have not been universally used in *Talaromyces*, and relatively few sequences for these genes are available from the NCBI database. Currently, although phylogenetic trees of the genus *Talaromyces* constructed from these remain imprecise, the genes have been used for identifying *Penicillium* species [6].

When constructing phylogenetic trees, it is necessary to delete redundant and irrelevant sequences. In the BLAST comparison results, the sequences related to some species did not include the corresponding type strains. In such cases, the sequence information should be validated, as the sequences might have been misidentified (wrongly identified as another species). For example, in Table 1, species marked with a yellow background color did not cluster with the type strains of the corresponding species, and phylogenetic results indicate that these species may be new species—*Talaromyces_stollii* (blue font) (Appendix A). This discrepancy is due to the fact that not all sequences in the NCBI database have been verified. Therefore, type strains of these species should be added as references for molecular identification and construction of phylogenetic trees. Here, we selected sequences from the type strain of *T. pinophilus* and other related strains, and some sequences of *T. pinophilus* that were not relevant to our study were removed.

In addition, when building phylogenetic trees, if the sequences used to construct the tree are not sufficiently comprehensive, the strain to be identified will only cluster with the sequences of similar species, rather than the sequence of the closest species. This problem occurs because the sequences closest to that of the strain to be identified at the genetic level may not be included in the NCBI-BLAST results due to differences in the length of the uploaded sequences or differences in gene coverage, resulting in an inaccurate phylogenetic tree. Specifically, analysis of NCBI-BLAST results revealed that most sequences included only partial sequences of a gene (not all the bases of the gene). The uploaded gene sequences are inconsistent in length, and each sequence contains a different region of the full-length gene. These differences result in the common phenomenon of sequences that appear most similar in the alignment results not being those that are actually most similar to the destination sequence (i.e., the results are inaccurate).

In summary, in previous international research on filamentous fungal species such as *Penicillium* and *Talaromyces*, the standard research method (GCPSR) was recommended. This polyphasic approach, which involved multigene phylogeny, morphological descriptions using macro-morphological and micro-morphological characters. To build an accurate phylogenetic tree based on NCBI-BLAST sequences, it is essential to refer to sequences provided in the authoritative literature. For the gene sequences of type strains, the selected sequences should be validated or verified. Using ITS and four specific gene sequences in various *Talaromyces* species, we constructed two phylogenetic trees (tree 1: ITS, *BenA*, *CaM*, *RPB1*, and *RPB2* (Figure 1); tree 2: ITS, *BenA*, *CaM*, and *RPB2* (Figure 2)) based on combinations of multiple genes. In the genus *Talaromyces*, combinations of three or four genes are more common, whereas analyses of five genes have been rare. At present, ITS, *BenA*, *CaM*, *RPB1*, and *RPB2* are the most authoritative and reliable genes for the identification of *Talaromyces* species. The preliminary phylogenetic tree construction results indicate that the species most closely related to strain JP-NJ4 is *T. liani*. The concatenated phylogenies of five (or four) gene regions and single gene phylogenetic tree (*BenA*, *RPB1*, and *RPB2* genes) all also show that *T. nanjingensis* strain JP-NJ4 and *T. liani* clustered together but differ markedly in their genetic distance from type strain of *T*. *liani* and other multiple collections of *T*. *liani*. The morphology of JP-NJ4 (M 2012167) largely matches the characteristics of *T. liani*, but the rich and specific morphological information provided by its colonies was different from that of *T. liani*. In addition, strain JP-NJ4 could produce reduced conidiophores with solitary phialides. From molecular and phenotypic data, strain JP-NJ4 was identified as a putative novel *Talaromyces* fungal species, designated *T. nanjingensis*. *T. nanjingensis* also can produce yellow, orange, and red soluble pigments in their mycelium, including diffusing pigments, similar to other species of the genus [81,82]. Due to the rich and specific morphological information provided by colonies, additional colony morphology photographs of this strain growing at 25 °C for 14 days on 10 different media were captured. We believe that it is essential to apply this information as part of the general method of strain identification. Future research will focus on the ecological function of *T. nanjingensis* JP-NJ4 and its impacts on the environment in terms of ecological security will also be assessed.


**The information of the culture preservation institutions involved is as follows (alphabetically):**
**ACCC**: Agricultural Culture Collection of China.**ATCC**: American Type Culture Collection, Manassas, VA, USA (WDCM 1) http://www.atcc.org/, accessed on 18 January 2022;**CABI**: Centre for Agriculture and Bioscience International (International Mycological Institute, CABI Genetic Resource Collection).**CBS**: culture collection of the CBS-KNAW Fungal Biodiversity Centre, Utrecht, Netherlands (WDCM 133) http://www.cbs.knaw.nl/databases/index.htm, accessed on 18 January 2022.**DTO**: internal culture collection of CBS-KNAW Fungal Biodiversity Centre; IMI, CABI Genetic Resources Collection, Surrey, UK (WDCM 214) http://www.cabi.org/, accessed on 18 January 2022.**FERM**: (Patent and Bio-Resource Center, National Institute of Advanced Industrial Science and Technology-AIST).**FMR**: facultad de medicina, Universidad de Oviedo. 33071-Oviedo. Spain. Institute de Investigaciones Biomidicas C.S.I.C., Facultad de Medicina UAM, E-28029 Madrid, Spain.**HMAS**: Fungarium of Institute of Microbiology.**IBT**: culture collection of Center for Microbial Biotechnology (CMB) at Department of Systems Biology, Technical University of Denmark (WDCM 758) http://www.biocentrum.dtu.dk/, accessed on 18 January 2022.**MUCL**: Mycotheque de l’Universite catholique de Louvain, Leuven, Belgium (WDCM 308).**NBRC**: Biological Resource Center, NITE.**NRRL**: ARS Culture Collection, U.S. Department of Agriculture, Peoria, Illinois, USA (WDCM 97) http://nrrl.ncaur.usda.gov/, accessed on 18 January 2022.


## Figures and Tables

**Figure 1 jof-08-00155-f001:**
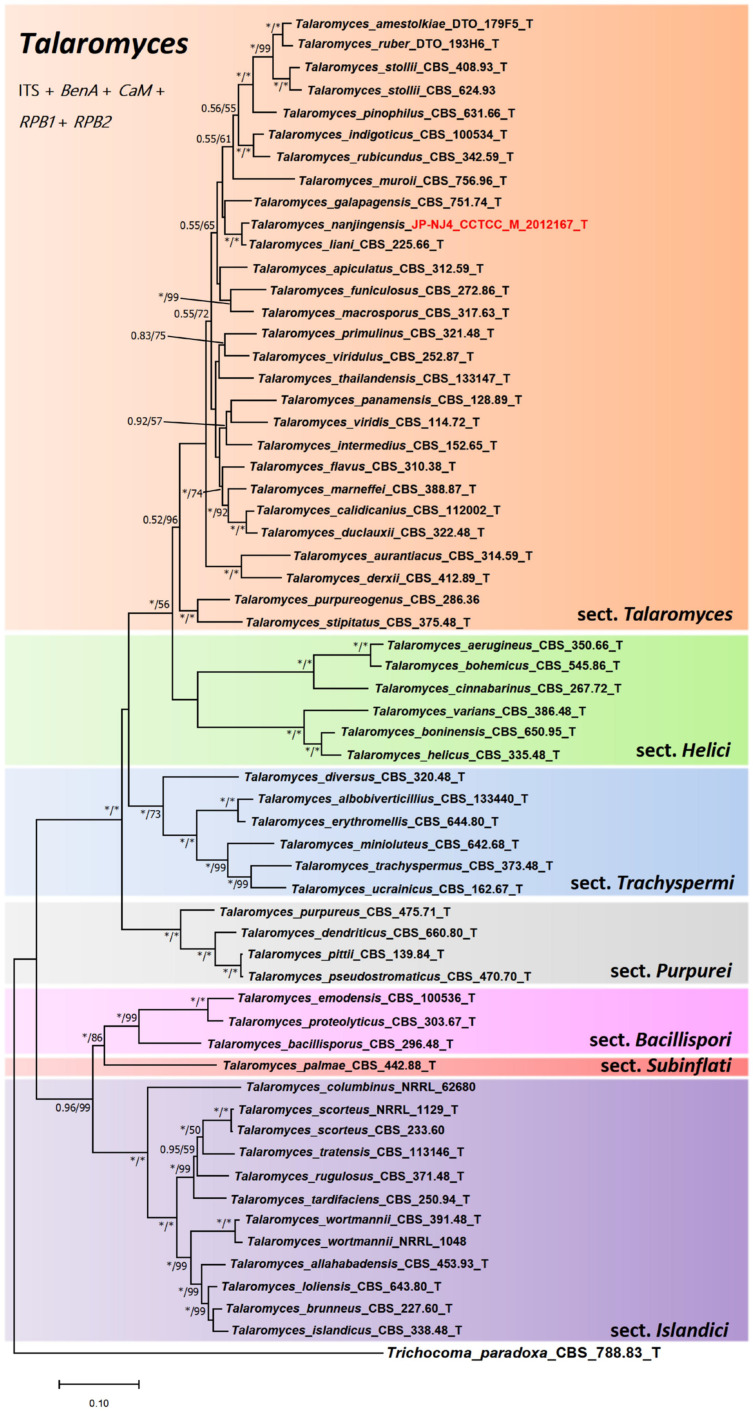
Combined phylogeny of the ITS, *BenA*, *CaM*, *RPB1*, and *RPB2* gene regions of species from *Talaromyces*. Maximum likelihood tree of strain JP-NJ4 was constructed. *Trichocoma paradoxa* (CBS_788.83_T) was chosen as out-group. Support in nodes is indicated above branches and is represented by posterior probabilities (BI analysis) and bootstrap values (ML analysis). Full support (1.00/100%) is indicated with an asterisk (*). Bootstrap values lower than 50 is hidden. Best-fit model of Bayesian Inference phylogeny according to BIC: SYM+I+G4; best-fit model of Maximum likelihood phylogeny according to AIC: Kimura 2-parameter (K2) +G+I; alignment, 444 (ITS) + 294 (*BenA*) + 489 (*CaM*)+ 491 (*RPB1*) + 677 (*RPB2*) = 2395 bp. Scale bar: 0.10 substitutions per nucleotide position. T indicates ex type. The strain with red font is the strain JP-NJ4 to be identified.

**Figure 2 jof-08-00155-f002:**
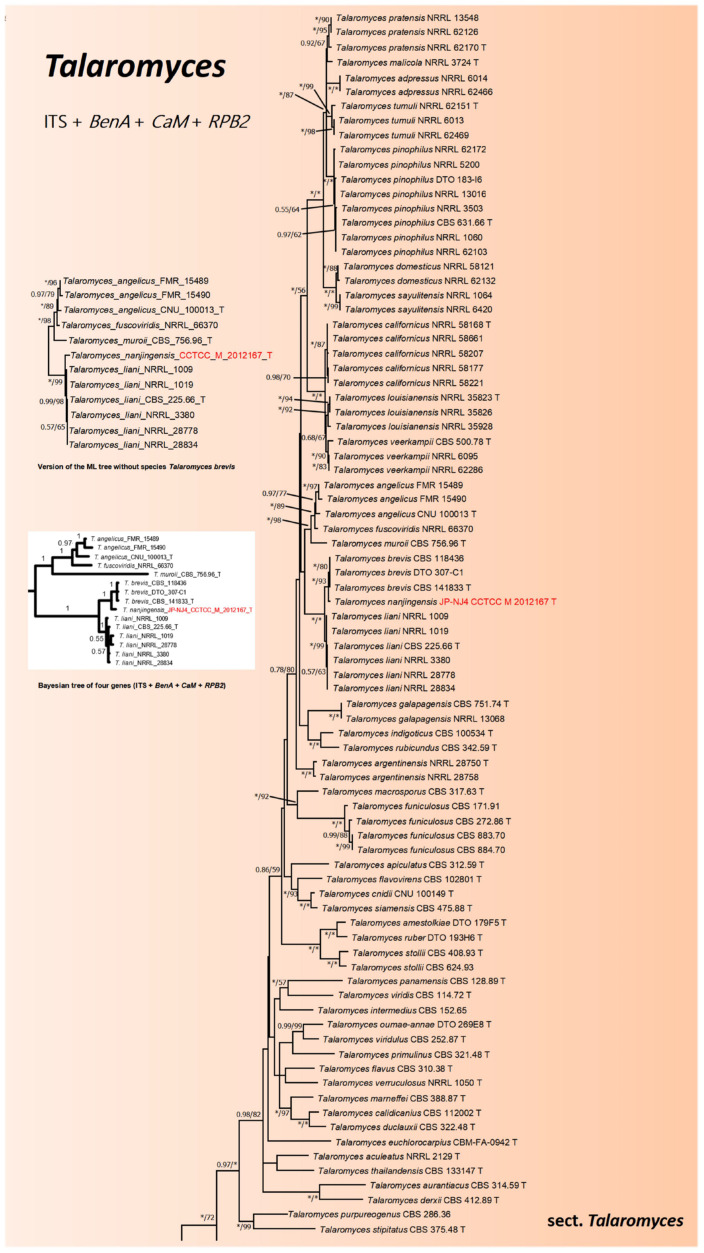
Combined phylogeny of the ITS, *BenA*, *CaM*, and *RPB2* gene regions of species from *Talaromyces*. Maximum likelihood tree of strain JP-NJ4 was constructed. *Trichocoma paradoxa* (CBS_788.83_T) was chosen as out-group. Support in nodes is indicated above branches and is represented by posterior probabilities (BI analysis) and bootstrap values (ML analysis). Full support (1.00/100%) is indicated with an asterisk (*). Bootstrap values lower than 50 is hidden. Best-fit model of Bayesian Inference phylogeny according to BIC: SYM+I+G4; best-fit model of Maximum likelihood phylogeny according to AIC: Kimura 2-parameter (K2) +G+I; alignment, 439 (ITS) + 284 (*BenA*) + 482 (*CaM*) + 677 (*RPB2*) = 1882 bp. Scale bar: 0.05 substitutions per nucleotide position. T indicates ex type. The strain with red font is the strain JP-NJ4 to be identified.

**Figure 3 jof-08-00155-f003:**
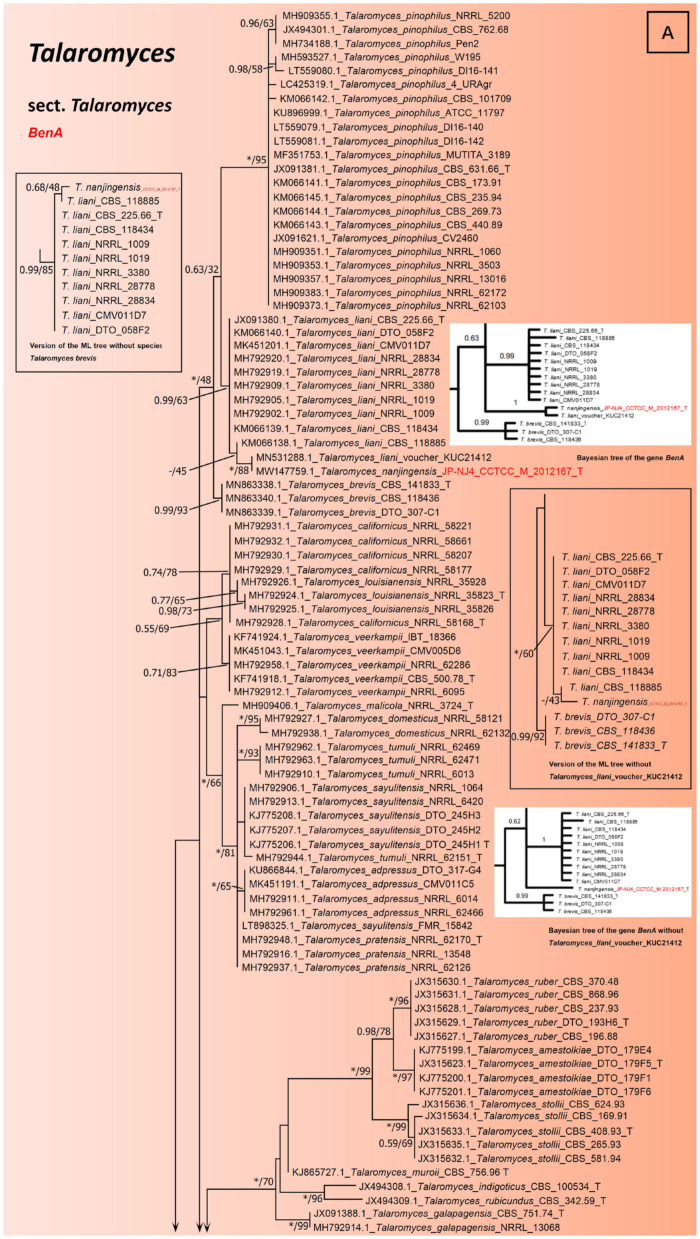
Maximum likelihood phylogeny of *BenA* gene regions for strain JP-NJ4 and other species classified in *Talaromyces* sect. *Talaromyces*. (**A**) Short sequence version with multiple species, alignment, *BenA* 316 bp. Best-fit model of Bayesian Inference phylogeny according to BIC: K80 (K2P) +I+G4; best-fit model of Maximum likelihood phylogeny according to AIC: Kimura 2-parameter (K2) +G; (**B**) Long sequence version with few species, alignment, *BenA* 391 bp. Best-fit model of Bayesian Inference phylogeny according to BIC: SYM+G4; best-fit model of Maximum likelihood phylogeny according to AIC: Kimura 2-parameter (K2) +G. *Talaromyces dendriticus* (CBS_660.80_T) was chosen as out-group. Support in nodes is indicated above branches and is represented by posterior probabilities (BI analysis) and bootstrap values (ML analysis). Full support (1.00/100%) is indicated with an asterisk (*). Missing data from the Bayesian tree are indicated with a dash (-). Scale bar: 0.05 substitutions per nucleotide position. T indicates ex type. The strain with red font is the strain JP-NJ4 to be identified.

**Figure 4 jof-08-00155-f004:**
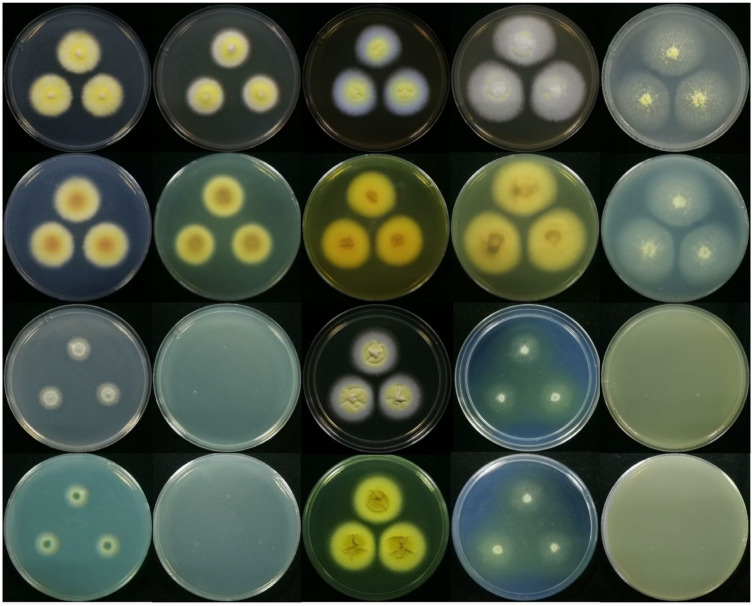
Macromorphological characters of strain JP-NJ4 (CCTCC M 2012167) (Inoculation at 25 °C for 7 days). Colonies from left to right: (the top two rows) CZ, CYA, MEA, MEAbl, OA, and the reverse side corresponding to these media; (the bottom two rows) DG18, CYAS, YES, CREA, HAY, and the reverse side corresponding to these media (the background color is black).

**Figure 5 jof-08-00155-f005:**
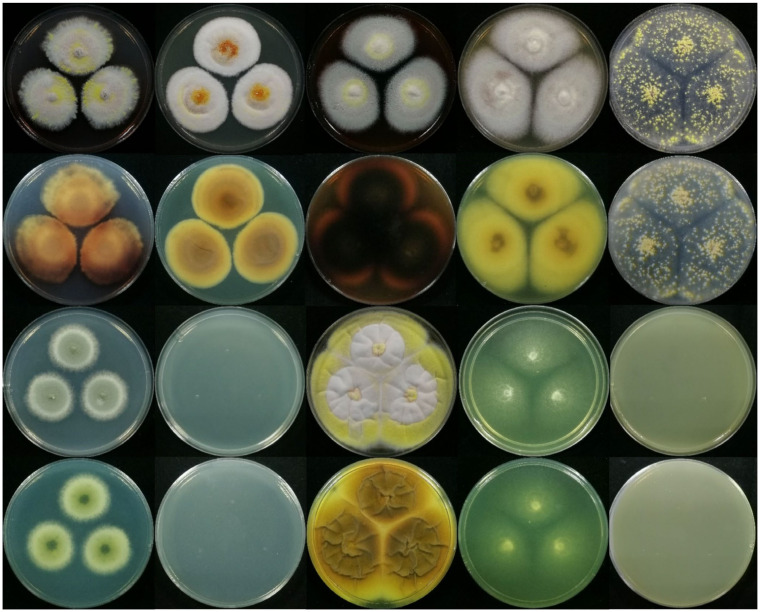
Macromorphological characters of strain JP-NJ4 (Inoculation at 25 °C for 14 days). Colonies from left to right: (the top two rows) CZ, CYA, MEA, MEAbl, OA, and the reverse side corresponding to these media; (the bottom two rows) DG18, CYAS, YES, CREA, HAY, and the reverse side corresponding to these media (the background color is black).

**Figure 6 jof-08-00155-f006:**
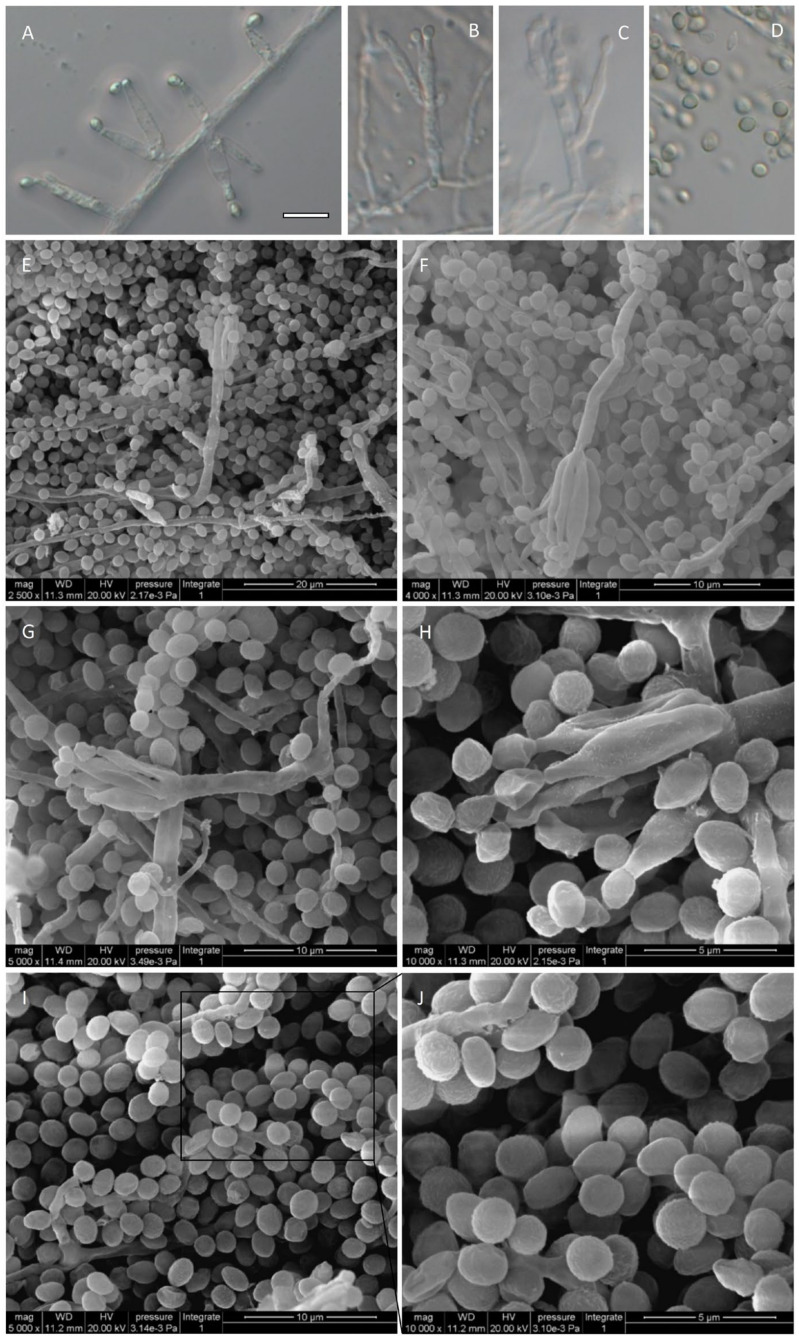
Micromorphological characters of JP-NJ4 (CCTCC M 2012167) (anamorphic stage) (inoculation for 1–2 wk on MEA). (**A**–**D**) Conidiophores and conidia, observed by optical microscope (Zeiss). (**A**) Reduced conidiophores consisting of solitary phialides. (**B**) Monoverticillate conidiophores. (**C**) Biverticillate conidiophores. (**D**) Conidia. (**E**–**J**) Conidiophores and conidia, observed by scanning electron microscope (SEM). (**E**–**G**) Conidiophores and conidia at different magnification (E. 2500×; F. 4000×; G. 5000×). (**H**) Phialides and conidia (10,000×). (**I**–**J**) Conidia. Scale bars: A = 10 μm, applies to A–D. E = 20 μm; F = 10 μm; G = 10 μm; H = 5 μm; I = 10 μm; J = 5 μm.

**Figure 7 jof-08-00155-f007:**
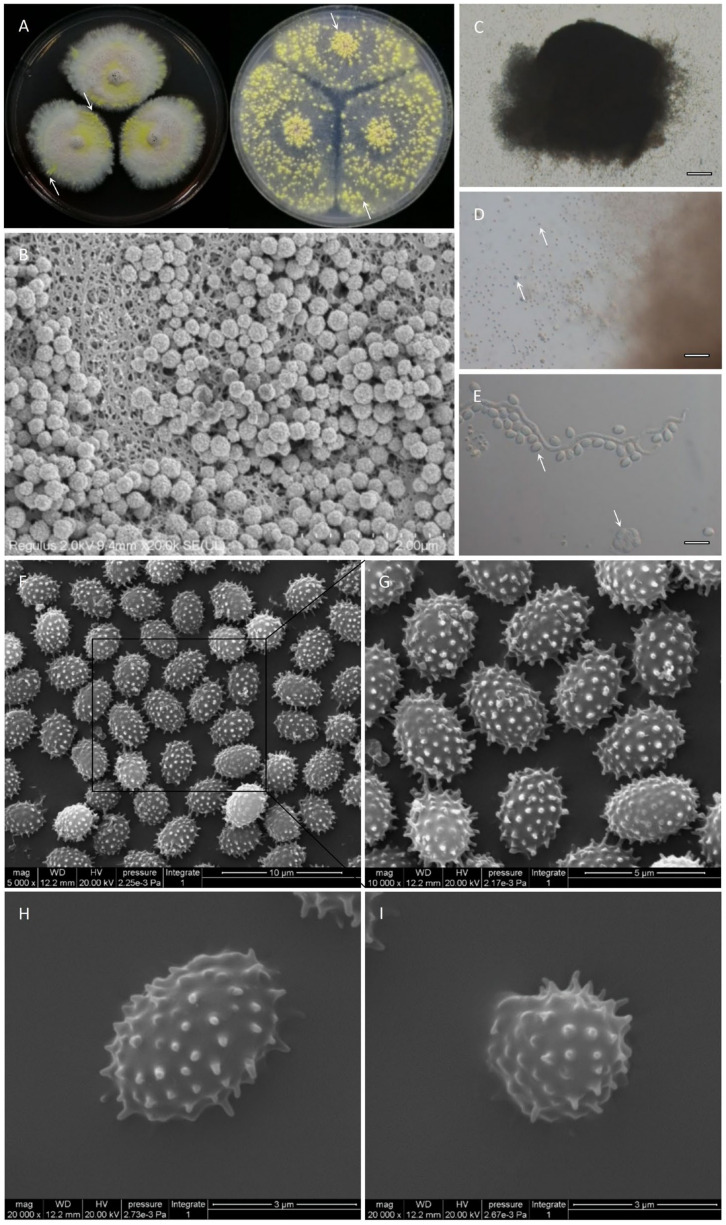
Teleomorphic stage of JP-NJ4 (CCTCC M 2012167). (**A**) Colonies inoculated for 2 wk on CZ (left) and OA (right). (**B**–**I**) Micromorphological characters of JP-NJ4. (**B**) Primary ascomata collected from OA (inoculation for 1 wk), observed by scanning electron microscope (SEM). (**C**–**D**) A mature ascoma that is releasing asci and ascospores at different magnification ((**C**) 5×; (**D**) 20×), observed by optical microscope (Zeiss). (**E**) An ascus and ascospores (100×). (**F**–**I**) Ascospores, observed by SEM. Scale bars: B = 2 μm; C = 200 μm; D = 50 μm; E = 10 μm; F = 10 μm; G = 5 μm; H = 3 μm; I = 3 μm.

**Table 1 jof-08-00155-t001:** Collection numbers of strains, isolation details and GenBank accession numbers of the five genes/region used for phylogenetic analysis of the strain JP-NJ4.

Species Name	Collection Number	Substrate and Origin	GenBank Accession Number				
			ITS	*BenA*	*CaM*	*RPB1*	*RPB2*
**strain JP-NJ4**	M 2012167	Rhizosphere soil from *Pinus massoniana*; Nanjing, Jiangsu, China	MW130720	MW147759	MW147760	MW147761	MW147762
** *Talaromyces brevis* **	CBS 141833 (T) = DTO 349-E7	Soil; Beijing, China	MN864269	MN863338	MN863315		MN863328
	DTO 307-C1	Soil; Zonguldak, Turkey	MN864270	MN863339	MN863316		MN863329
	CBS 118436 = DTO 004-D8	Soil; Maroc	MN864271	MN863340	MN863317		MN863330
** *Talaromyces liani* **	CBS 225.66 (T)	Soil; China	JN899395	JX091380	KJ885257	JN680280	KX961277
	CBS 118434	Soil in orchid garden; Sanur, Bali, Indonesia	KM066208	KM066139	MK451683= KP453744	-	-
	CBS 118885	Soil of pepper field; DaeJeon, Korea	KM066210	KM066138	-	-	-
	NRRL 1009	Derived from Biourge 368	MH793030	MH792902	MH792966	-	MH793093
	NRRL 1014	= 1009	MH793031	MH792903	MH792967	-	MH793094
	NRRL 1015	= 1009	MH793032	MH792904	MH792968	-	MH793095
	NRRL 1019	USA, Arizona, isol ignotae, *KD Butler*, 1936.	MH793033	MH792905	MH792969	-	MH793096
	NRRL 3380	China, isol ex soil, = CBS 225.66	MH793037	MH792909	MH792973	-	MH793100
	NRRL 28778	Brazil, isol ex soil, *RW Jackson*, 1956.	MH793047	MH792919	MH792983	-	MH793110
	NRRL 28834	India, isol ignotae	MH793048	MH792920	MH792984	-	MH793111
	CMV011D7	*Passiflora edulis*; South Africa	-	MK451201	-	-	-
	KUC21412	Mudflat; South Korea	MN518409	MN531288	-	-	-
	DTO 058F2	Heat tretaed corn kernels; the Netherlands	KM066209	KM066140	-	-	-
** *Talaromyces aculeatus* **	CBS 289.48 (T) = NRRL2129	Textile; USA	KF741995	KF741929	KF741975 = JX140684 = MH792972	-	KM023271
	CBS 282.92	Soil in secondary forest; Brazil	KF741981	KF741914	KF741946	-	-
	CBS 290.65	Nut; South Africa	KF741982	KF741915	KF741948	-	-
	CBS 563.92	Stem of *Dicymbe Altsonii*; French Guiana	KF741986	KF741920	KF741963	-	-
	CBS 136673 = IBT14255	Weathering wood stakes; Palmerston North, New Zealand	KF741990	KF741927	KF741970	-	-
** *Talaromyces adpressus* **	NRRL 6014	Peanuts; Unknown	MH793039	MH792911	MH792975	-	MH793102
	NRRL 62466	Peanuts; Unknown	MH793088	MH792961	MH793025	-	MH793152
	CBS 140620	Indoor air; China	KU866657	-	-	-	KU867001
	DTO 317-G4	Indoor air; China	-	KU866844	KU866741	-	-
	CMV011C5	Soil; South Africa	MK450741	MK451191	MK451673	-	-
** *Talaromyces aerugineus* **	CBS 350.66 (T)	Debris; United Kingdom	AY753346 = NR 147420	KJ865736	KJ885285	JN121657	JN121502
** *Talaromyces albobiverticillius* **	CBS 133440 (T) = *Penicillium albobiverticillium* isolate 900890701	Decaying leaves of a broad-leaved tree; Taiwan	HQ605705 = KF114734	KF114778	KJ885258	KF114753	KM023310
	CBS 133441	Decaying leaves of a broad-leaved tree; Taiwan	KF114733	KF114777	-	KF114755	-
** *Talaromyces allahabadensis* **	CBS 453.93 (T)	Cultivated soil; Allahabad, India	KF984873	KF984614 = JX494298	KF984768	JN680309	KF985006
	CBS 178.81	*Crepis zacintha*; Alicante, Spain; Type of *Penicillium zacinthae*	KF984863	KF984612	KF984767	-	KF985004
	CBS 441.89	Seed groud; Denmark	KF984872	KF984613	KF984759	-	KF985005
	CBS 137397 = DTO245E3	House dust; Mexico	KF984864	KF984605	KF984761	-	KF984998
	CBS 137399 = DTO267H6	House dust; Thailand	KF984866	KF984607	KF984762	-	KF984997
** *Talaromyces amestolkiae* **	CBS 132696 (T) = DTO179F5	House dust; South Africa	JX315660 = NR 120179	JX315623	KF741937 = JX315650	JX315679	JX315698
	DTO179E4	House dust; South Africa	KJ775706	KJ775199	JX140685	-	-
	DTO179F1	House dust; South Africa	KJ775707	KJ775200	JX140686	-	-
	DTO179F6	House dust; South Africa	KJ775708	KJ775201	-	-	-
** *Talaromyces angelicus* **	KACC 46611 (T) = CNU 100013 = DTO303E2	Dried roots of *Angelica gigas*; Pyeongchang, Korea	KF183638	KF183640	KJ885259	-	KX961275
	FMR 15489	Unknown	LT899791	LT898316	LT899773	-	LT899809
	FMR 15490	Unknown	LT899792	LT898317	LT899774	-	LT899810
** *Talaromyces apiculatus* **	CBS 312.59 (T)	Soil; Japan	JN899375 = NR 121530	KF741916 = JX091378	KF741950	JN680293	KM023287
	CBS 548.73	Soil; Suriname	KF741985	KF741919	KF741962	-	-
	CBS 101366	Soil; Hong Kong, China	KF741977	KF741910	KF741932	-	-
** *Talaromyces argentinensis* **	NRRL 28750 (T)	Soil; Unknown	MH793045 = NR 165525	MH792917	MH792981	-	MH793108
	NRRL 28758	Soil; Unknown	MH793046	MH792918	MH792982	-	MH793109
** *Talaromyces assiutensis* **	CBS 147.78 (T)	Soil; Egypt	JN899323	KJ865720	KJ885260	JN680275	KM023305
	CBS 645.80	Gossypium; India; Type of *Talaromyces gossypii*	JN899334= NR 147423	KF114802	-	JN680317	-
	CBS 116554	Pasteurised canned strawberries; the Netherlands	KM066167	KM066124	MK451674	-	-
	CBS 118440	Soil; Fes, Marocco	KM066168	KM066125	MK451675	-	-
** *Talaromyces atricola* **	CBS 255.31 (T)	Unknown	KF984859	KF984566	KF984719	-	KF984948
** *Talaromyces atroroseus* **	CBS 133442 (T)	House dust; South Africa	KF114747 = NR 137815	KF114789	KJ775418	KF114763	KM023288
	DTO267I1	House dust; Thailand	KJ775716	KJ775209	-	-	-
	DTO270D5	House dust; Mexico	KJ775734	KJ775227	-	-	-
	DTO270D6	House dust; Mexico	KJ775735	KJ775228	-	-	-
** *Talaromyces aurantiacus* **	CBS 314.59 (T)	Soil; Georgia	JN899380 = NR103681.2	KF741917	KF741951	JN680294	KX961285
** *Talaromyces australis* **	IBT14256 (T)	Unknown	KF741991 = NR 147431	KF741922	KF741971	-	-
	IBT14254	Unknown	KF741989	KF741923	KF741969	-	-
	MDL18159	Bronchoscopy; USA	MK601840	MK626507	-	MK626517	-
** *Talaromyces austrocalifornicus* **	CBS 644.95 (T)	Soil; California, USA	JN899357 = NR 137079	KJ865732	KJ885261	JN680316	-
** *Talaromyces bacillisporus* **	CBS 296.48 (T)	Leaf; New York, USA	JN899329	AY753368	KJ885262	JN121634	JF417425
	CBS 102389	Sludge of anaerobic pasteurised organic household waste; Sweden	KM066179	KM066135	-	-	-
	CBS 110774	Rye bread; the Netherlands	KM066180	KM066136	-	-	-
	CBS 116927	Soil; the Netherlands	KM066181	KM066137	-	-	-
** *Talaromyces bohemicus* **	CBS 545.86 (T)	Peloids for balneological purposes; Czech Republic	JN899400 = NR 137081	KJ865719	KJ885286	JN121699	JN121532
** *Talaromyces boninensis* **	CBS 650.95 (T)	Peloids for balneological purposes; Czech Republic	JN899356 = NR 145157	KJ865721	KJ885263	JN680319	KM023276
** *Talaromyces brunneus* **	CBS 227.60 (T)	Milled rice imported into Japan; Thailand	JN899365 = NR 111688	KJ865722 = JX494296	KJ885264	JN680281	KM023272
** *Talaromyces calidicanius* **	CBS 112002 (T)	Soil; Nantou County, Taiwan	JN899319 = HQ149324 = NR 103665.2	HQ156944	KF741934 = JX140688	JN899305	KM023311
	ACCC:39162	Luffa; Beijing; China	KY225703	KY225714	-	KY225712	-
	ACCC:39164	Cucumber; Beijing; China	KY225702	KY225715	-	KY225711	-
** *Talaromyces californicus* **	NRRL 58168 (T)	Air sample; Unknown	MH793056 = NR 165527	MH792928	MH792992	-	MH793119
	NRRL 58177	Air sample; Unknown	MH793057	MH792929	MH792993	-	MH793120
	NRRL 58207	Air sample; Unknown	MH793058	MH792930	MH792994	-	MH793121
	NRRL 58221	Air sample; Unknown	MH793059	MH792931	MH792995	-	MH793122
	NRRL 58661	Air sample; Unknown	MH793060	MH792932	MH792996	-	MH793123
** *Talaromyces cecidicola* **	CBS 101419 (T) = *Penicillium cecidicola* strain DAOM 233329 = *Penicillium cecidicola* isolate KAS504	Cynipid insect galls on *Quercus pacifica* twigs; Oregon, USA	AY787844 = MH862736	FJ753295	KJ885287	-	KM023309
** *Talaromyces* ** ** *cellulolyticus* **	Y-94 = FERM: BP-5826	Unknown; A synonym of *Talaromyces pinophilus*	AB474749	AB773823	-	AB856422	-
** *Talaromyces chloroloma* **	DAOM 241016 (T) = *Penicillium* sp. CMV-2008a isolate Pen389 = *Penicillium* sp. CMV-2008a isolate CV389	Fynbos soil; Western Cape, South Africa	FJ160273	GU385736	KJ885265	-	KM023304
	DTO 180-F4 = *Penicillium* sp. CMV-2008a isolate CV390 = *Penicillium* sp. CMV-2008a isolate Pen390	Fynbos soil; South Africa	FJ160272	GU385737	-	-	-
	DTO 182-A5 = CV785 = CV0785	Air sample; Malmesbury, South Africa	JX091485	JX091597	JX140689	-	MK450871
** *Talaromyces cinnabarinus* **	CBS 267.72 (T)	Soil, Japan	JN899376	AY753377	KJ885256	JN121625	JN121477
	CBS 357.72	Soil, Japan	KM066178 = MH860496 = AY753347	KM066134 = AY753376	-	-	-
** *Talaromyces* ** ** *cnidii* **	KACC 46617 (T) = DTO 303-E1 = CNU 100149	Dried roots of *Cnidium officinale*; Jecheon, Korea	KF183639	KF183641	KJ885266	-	KM023299
	DTO 269-H8	House dust; Thailand	KJ775724	KJ775217	KJ775426	-	-
	DTO 270-A4	House dust; Thailand	KJ775729	KJ775222	KJ775430	-	-
	DTO 270-A8	House dust; Thailand	KJ775730	KJ775223	KJ775431	-	-
	DTO 270-B7	House dust; Thailand	KJ775731	KJ775224	KJ775432	-	-
** *Talaromyces coalescens* **	CBS 103.83 (T)	Soil under *Pinus* sp.; Spain	JN899366 = NR 120008	JX091390	KJ885267	-	KM023277
** *Talaromyces columbinus* **	NRRL 58811 (T)	Air; Loisiana, USA	KJ865739 = NR 147433	KF196843	KJ885288	-	KM023270
	CBS 137393 = DTO 189-A5	Chicken feed (Unga); Nairobi, Kenya	KF984794	KF984659	KF984671	-	KF984897
	NRRL 58644	Air; Maryland, USA	KF196899	KF196842	KF196880	-	KF196987
	NRRL 62680	Corn grits; Illinois, USA	KF196901	KF196844	KF196882	KF196949	KF196988
** *Talaromyces convolutus* **	CBS 100537 (T)	Soil; Kathmandu, Nepal	JN899330 = NR 137157	KF114773	-	JN121553	JN121414
** *Talaromyces dendriticus* **	CBS 660.80 (T)	*Eucalyptus pauciflora* leaf litter; New South Wales, Australia	JN899339	JX091391	KF741965	JN121714	KM023286= JN121547
	DAOM 226674 = *Penicillium dendriticum* isolate KAS849	*Doryanthes excelsa* spathes; Mangrove Mountain, New South Wales, Australia	AY787842	FJ753293	-	-	-
	DAOM 233861 = *Penicillium dendriticum* isolate KAS1190	Unindentified insect gall on *Eucalyptus* leaf; Kalnura, New South Wales, Australia	AY787843	FJ753294	-	-	-
	DTO 183-G3 = CV2026	Mite; Struisbaai, South Africa	JX091486	JX091619	JX140692	-	MK450872
** *Talaromyces* ** ** *derxii* **	CBS 412.89 (T)	Cultivated soil; Japan	JN899327 = NR 145152	JX494306	KF741959	JN680306	KM023282
** *Talaromyces* ** ** *diversus* **	CBS 320.48 (T)	Leather; USA	KJ865740	KJ865723	KJ885268	JN680297	KM023285
	DTO 133-A7	House dust; Thailand	KJ775701	KJ775194	-	-	-
	DTO 133-E4	House dust; Thailand	KJ775702	KJ775195	-	-	-
	DTO 133-I6	Lotus tea; produced in Vietnam, imported to the Netherlands	KJ775700	KJ775193	-	-	-
	DTO 244-E6	House dust; New Zealand	KJ775712	KJ775205	-	-	-
** *Talaromyces domesticus* **	NRRL 58121	Floor swab; Unknown	MH793055	MH792927	MH792991	-	MH793118
	NRRL 62132	Exposed cloth; Unknown	MH793066	MH792938	MH793002	-	MH793129
** *Talaromyces duclauxii* **	CBS 322.48 (T)	Canvas; France	JN899342 = NR 121526	JX091384	KF741955	JN121643	JN121491
** *Talaromyces* ** ** *emodensis* **	CBS 100536 (T)	Soil; Kathmandu, Nepal	JN899337 = NR 137077	KJ865724	KJ885269	JN121552	JF417445
** *Talaromyces* ** ** *erythromellis* **	CBS 644.80 (T)	Soil from creek bank; New South Wales	JN899383	HQ156945	KJ885270	JN680315	KM023290
** *Talaromyces euchlorocarpius* **	PF 1203 (T) = DTO 176I3 = CBM-FA-0942	Soil; Yokohama, Japan	AB176617	KJ865733	KJ885271	-	KM023303
** *Talaromyces flavovirens* **	CBS 102801 (T)	Dead leaves of *Quercus ilex*; Parque del Retiro, Madrid, Spain	JN899392	JX091376	KF741933	-	KX961283
	DAOM236381	Leaves of *Quercus suber*; port de la Selva, Girona, Spain	JX013912	JX091373	-	-	-
	DAOM236382	Leaves of *Quercus suber*; Selva de Mar, Girona, Spain	JX013913	JX091374	-	-	-
	DAOM236383	Leaves of *Quercus suber*; Barraca d’en Rabert, Paau, Girona, Spain	JX013914	JX091377	-	-	-
	DAOM236384	Leaves of *Quercus suber*; Xovar, Alt Palacia, Valencia	JX013915	JX091375	-	-	-
** *Talaromyces* ** ** *flavus* **	CBS 310.38 (T)	Unknown; New Zealand	JN899360	JX494302	KF741949 = FJ530982	JN121639	JF417426
	CBS 437.62	Compost; Bonn, Germany	KM066202	KM066156	-	-	-
** *Talaromyces* ** ** *francoae* **	CBS 113134 (T)	Leaf litter; Colombia	NR 154940	-	-	-	-
	DTO 056D9	Leaf litter; Colombia	KX011510	KX011489	KX011501	-	-
** *Talaromyces funiculosus* **	CBS 272.86 (T)	*Lagenaria vulgaris*; India	JN899377 = NR 103678.2	JX091383	KF741945	JN680288	KM023293
	CBS 171.91	Unknown	KM066193	KM066162	MK451679	-	MK450873
	CBS 883.70	Unknown; Java	KM066196	KM066163	MK451680	-	MK450874
	CBS 884.70	Unknown; Java	KM066195	KM066164	MK451681	-	MK450875
	CBS 885.71	Air; Java, Jakarta	KM066194	KM066165	-	-	MK450876
** *Talaromyces* ** ** *fuscoviridis* **	CBS 193.69 (T)	Unknown	KF741979 = NR 153227	KF741912	KF741942	-	-
	NRRL 66370	Unknown	MH793092	MH792965	MH793029	-	MH793156
** *Talaromyces galapagensis* **	CBS 751.74 (T)	Shaded soil under *Maytenus obovate*; Galapagos Islands, Isla, Santa Cruz, Ecuador	JN899358 = NR 147426	JX091388 = KF114770	KF741966	JN680321	KX961280
	NRRL 13068	*Maytenus obovata*	MH793042	MH792914	MH792978	-	MH793105
** *Talaromyces hachijoensis* **	IFM 53624 (T) = PF 1174 = CBM-FA-0948	Soil; Hachijojima, Japan	AB176620	-	-	-	-
** *Talaromyces* ** ** *helicus* **	CBS 335.48 (T)	Soil; Sweden	JN899359 = NR 147427	KJ865725	KJ885289	JN680300	KM023273
	CBS 134.67	Green house soil under *Lycopersicon esculentum*; Wageningen, the Netherlands	KM066176	KM066133	-	-	-
	CBS 550.72	Saline soil; Vallee de la Seille, France	KM066177 = MH860565	KM066132	-	-	-
	CBS 649.95 = *Talaromyces barcinensis*	Unknown	JN899349 = MH862547 = NR 137078	KJ865737	-	JN680318	-
	CBS 652.66	Unknown	JN899335	KJ865738	-	JN680320	-
** *Talaromyces indigoticus* **	CBS 100534 (T)	Soil; Japan	JN899331 = NR 137076	JX494308	KF741931	JN680323	KX961278
** *Talaromyces intermedius* **	CBS 152.65 (T)	Allauvial pasture and swamp soil; Nottingham, England	JN899332 = NR 145154	JX091387	KJ885290	JN680276	KX961282
** *Talaromyces* ** ** *islandicus* **	CBS 338.48 (T)	Unknown; Cape Town, South Africa	KF984885	KF984655 = JX494293	KF984780	JN121648	KF985018 = JN121495
	CBS 165.81	spice mixture used in sausage making industry; Spain; Type of *Penicillium aurantioflammiferum*	KF984883	KF984653	KF984778	-	KF985016
	CBS 394.50	Kapok fibre; unkown	KF984886	KF984656	KF984781	-	KF985019
	CBS 117284	Wheat flour; the Netherlands	KF984882	KF984652	KF984777	-	KF985015
** *Talaromyces* ** ** *kabodanensis* **	DI16-149	Unknown	-	-	LT795598	-	LT795599
** *Talaromyces* ** ** *kendrickii* **	IBT13593 (T)	Unknown	KF741987 = NR 147430	KF741921	KF741967	-	-
	IBT14128	Unknown	KF741988	KF741925	KF741968	-	-
	CBS 100105	Unknown	KF741976	KF741909	KF741930	-	-
	CBS 133088	Unknown	KF741978	KF741911	KF741939	-	-
** *Talaromyces loliensis* **	CBS 643.80 (T)	Rye grass (*Lolium*); New Zealand	KF984888	KF984658	KF984783	JN680314	KF985021
	CBS 172.91	Soil; New Zealand	KF984887	KF984657	KF984782	-	KF985020
** *Talaromyces louisianensis* **	NRRL 35823 (T)	Air sample; Unknown	MH793052= NR 165526	MH792924	MH792988	-	MH793115
	NRRL 35826	Air sample; Unknown	MH793053	MH792925	MH792989	-	MH793116
	NRRL 35928	Air sample; Unknown	MH793054	MH792926	MH792990	-	MH793117
** *Talaromyces macrosporus* **	CBS 317.63 (T)	Apple juice; Stellenbosch, South Africa	JN899333 = NR 145155	JX091382	KF741952	JN680296	KM023292
	CBS 117.72	Cotton fabric; USA	KM066188	KM066148	-	-	-
	CBS 131.87	Faecal pellet of grasshopper; Malaysia	KM066191	KM066147	-	-	-
	CBS 353.72	Tentage; New Guinea	KM066189	KM066149	-	-	-
	DTO 077-C5	Pine apple concentrate; the Netherlands	KM066192	KM066150	-	-	-
	DTO 105-C4	Unknown	KM066190	KM066146	-	-	-
	BCC 14364	Unknown	AY753345	AY753373	-	-	-
	AS3.6680	Unknown	-	-	AY678608	-	-
** *Talaromyces malicola* **	NRRL 3724 (T)	Soil under apple tree; Unknown	MH909513= NR 165531	MH909406	MH909459	-	MH909567
** *Talaromyces marneffei* **	CBS 388.87 (T)	Bamboo rat (*Rhizomys sinensis*); Vietnam	JN899344 = NR 103671.2	JX091389	KF741958	JN899298	KM023283
	CBS 108.89	Human (male); China	KM066187	KM066157	-	-	-
	CBS 122.89	Male AIDS patient after travel to Indonesia	KM066183	KM066161	-	-	-
	CBS 135.94	Haemoculture; Nonthaburi, Thailand	KM066184	KM066158	-	-	-
	CBS 549.77	Man spleen; unknown	KM066185	KM066159	-	-	-
	CBS 119456	Male blood; Thailand	KM066186	KM066160	-	-	-
** *Talaromyces mimosinus* **	CBS 659.80 (T)	Soil from creek bank, New South Wales	JN899338	KJ865726	KJ885272	JN899302	-
	NRRL 13069 = NRRL 13609 (*BenA*)	Unknown	KX946911	KX946880	KX946897	-	KX946926
** *Talaromyces minioluteus* **	CBS 642.68 (T)	Unknown	JN899346 = NR 121527	KF114799	KJ885273	JN121709	JF417443
	CBS 137.84 = *Penicillium samsonii* strain CBS137.84	Fruit, damaged by insect; Valladolid, Spain	KM066171	KF114798	-	JN680273	-
	CBS 270.35	*Zea mays*; Castle Rock, Virginia, USA; Type of *Penicillium purpurogenum* var. *rubrisclerotium*	KM066172	KM066129	-	JN680287	-
** *Talaromyces muroii* **	CBS 756.96 (T)	Soil; Taiwan	JN899351 = NR 103672.2	KJ865727	KJ885274	JN680322	KX961276
	CBS 261.55	*Clematis*; Boskoop, the Netherlands	KM066200	KM066153	-	-	-
	CBS 283.58	Jute potato bag, treated with copper oxide ammonia; unknown	KM066197	KM066151	-	-	-
	CBS 284.58	Unknown; the Netherlands	KM066199	KM066152	-	-	-
	CBS 351.61	Chicken crop; the Netherlands	KM066198	KM066155	-	-	-
	CBS 889.96	Dung of sheep; Papua New Guinea	KM066201	KM066154	-	-	-
** *Talaromyces oumae-annae* **	CBS 138208 (T) = DTO 269-E8	House dust; South Africa	KJ775720 = NR 147432	KJ775213	KJ775425	-	KX961281
	CBS 138207 = DTO 180-B4	House dust; South Africa	KJ775710	KJ775203	KJ775421	-	-
** *Talaromyces palmae* **	CBS 442.88 (T)	*Chrysalidocarpus lutescens* seed; Wageningen, the Netherlands	JN899396	HQ156947	KJ885291	JN680308	KM023300
** *Talaromyces panamensis* **	CBS 128.89 (T)	Soil; Barro Colorado Island, Panama	JN899362	HQ156948 = JX091386	KF741936 = JX140695	JN899291	KM023284
** *Talaromyces paucisporus* **	PF 1150 (T) = IFM 53616 = CBM-FA-0944	Soil; Aso-machi, Japan	AB176603	-	-	-	-
** *Talaromyces* ** ** *piceus* ** ** *=* ** ***Talaromyces piceae*?**	CBS 361.48 (T)	Unknown	KF984792	KF984668	KF984680	-	KF984899
	CBS 116872	Production plant; the Netherlands	KF984788	KF984660	KF984678	-	KF984903
	CBS 132063	Straw used in horse stable; the Netherlands	KF984789	KF984665	KF984674	-	KF984904
	CBS 137363 = DTO58D1	Pectin; unknown	KF984787	KF984664	KF984677	-	KF984902
	CBS 137377 = DTO178F3	House dust; South Africa	KF984784	KF984661	KF984676	-	KF984900
** *Talaromyces pinophilus* **	CBS 631.66 (T)	PVC; France	JN899382 = NR 111691	JX091381	KF741964	JN680313	KM023291
	CBS 173.91	Unknown; USA	KM066206	KM066141	-	-	-
	CBS 235.94	Unknown; USA	KM066204	KM066145	-	-	-
	CBS 269.73	Unknown; Germany	KM066207	KM066144	KM520392= MK451686	-	-
	CBS 440.89	*Zea mays*; India	KM066203	KM066143	-	-	-
	CBS 762.68	Rhizosphere; India; Type of *Penicillium korosum*	JN899347	JX494301	-	-	-
	CBS 101709	Soil; Japan	KM066205	KM066142	KM520391 = MK451685	-	-
	DTO183-I6 = CV2460	*Protea repens* infructescense; Struisbaai, South Africa	JX091488	JX091621	JX140697	-	MK450878
	NRRL 1060	Seed; Unknown	MH909460	MH909351	MH909407	-	MH909514
	NRRL 3503	Radio set; Unknown	MH909462	MH909353	MH909409	-	MH909516
	NRRL 5200	Unknown; Type of *Penicillium korosum*	MH909464	MH909355	MH909411	-	MH909518
	NRRL 13016	Dung ball; Unknown	MH909466	MH909357	MH909413	-	MH909520
	NRRL 62103	Canvas cloth; Unknown	MH909482	MH909373	MH909429	-	MH909535
	NRRL 62172	Wheat; Unknown	MH909492	MH909383	MH909439	-	MH909545
	ATCC 11797	Unknown	KU729085	KU896999	-	-	-
	CABI IMI114933	Unknown; France	KC962105	KC992266	-	-	-
** *Talaromyces* ** ** *pittii* **	CBS 139.84 (T)	Clay soil under poplar trees; Spain	JN899325 = NR 103667.2	KJ865728	KJ885275	JN680274	KM023297
** *Talaromyces pratensis* **	NRRL 62170 (T)	Unknown	MH793075 = NR 165529	MH792948	MH793012	-	MH793139
	NRRL 13548	Corn; Unknown	MH793044	MH792916	MH792980	-	MH793107
	NRRL 62126	River water; Unknown	MH793065	MH792937	MH793001	-	MH793128
** *Talaromyces primulinus* **	CBS 321.48 (T)	Unknown; USA	JN899317 = NR 145151	JX494305	KF741954	JN680298	KM023294
** *Talaromyces proteolyticus* **	CBS 303.67 (T)	Granite soil; Ukraine	JN899387 = NR 103685.2	KJ865729	KJ885276	JN680292	KM023301
** *Talaromyces pseudostromaticus* **	CBS 470.70 (T)	Feather of *Hylocichla fuscescens*; Minnesota, USA	JN899371	HQ156950	KJ885277	JN899300	KM023298
** *Talaromyces ptychoconidium* **	DAOM 241017 (T) = DTO 180-E7 = CV2808 = *Penicillium* sp. CMV-2008c isolate CV319 = *Penicillium* sp. CMV-2008c isolate Pen322	Fynbos soil; Malmesbury, South Africa	FJ160266	GU385733	JX140701	-	KM023278
	DTO 180-E9 = *Penicillium* sp. CMV-2008c isolate Pen319 = *Penicillium* sp. CMV-2008c isolate CV322	Fynbos soil; Malmesbury, South Africa	FJ160267	GU385734	-	-	MK450879
	DTO 180-F1 = *Penicillium* sp. CMV-2008c isolate CV323	Fynbos soil; Malmesbury, South Africa	GQ414762	GU385735	-	-	-
** *Talaromyces purpureus* **	CBS 475.71 (T)	Soil; France	JN899328 = NR 145153	GU385739	KJ885292	JN121687	JN121522
** *Talaromyces purpurogenus* **	CBS 286.36 (T)	Parasitic on a culture of *Aspergillus oryzae*; Japan	JN899372 = NR 121529	JX315639	KF741947 = JX315655	JN680271	JX315709
	CBS 184.27	Soil; Lousiana, USA	JX315665 = MH854924	JX315637	JX315658	JX315684 = JN680270	-
	CBS 122434	Unknown	JX315663	JX315640	JX315659	JX315682	-
	CBS 132707 = DTO189A1	Moulded field corn; Wisconsin, USA	JX315661	JX315638	JX315642	JX315680	-
** *Talaromyces rademirici* **	CBS 140.84 (T)	Air under willow tree; Valladolid, Spain	JN899386 = NR 103684.2	KJ865734	-	-	KM023302
** *Talaromyces radicus* **	CBS 100489 (T)	Root seadling; New South Wales	KF984878	KF984599	KF984773	-	KF985013
	CBS 100488	Wheat root; New South Wales	KF984877	KF984598	KF984772	-	KF985012
	CBS 100490	Wheat root; New South Wales	KF984879	KF984600	KF984774	-	KF985014
	CBS 137382 = DTO181D5	Fynbos soil; South Africa	KF984875	KF984602	KF984775	-	KF985009
	DTO181D4	Fynbos soil; South Africa	KF984880	KF984601	KF984770	-	KF985008
	DTO181D7	Fynbos soil; South Africa	KF984881	KF984603	KF984771	-	KF985010
** *Talaromyces ramulosus* **	DAOM 241660 (T) = CV2837 = CV113	Soil; Malmesbury, South Africa	EU795706	FJ753290	JX140711	-	KM023281
	DTO 181-E3 = CV314 = CV0314	Mite; Stellenbosch, South Africa	JX091494	JX091626	JX140706	-	-
	DTO 181-F6 = CV394 = CV0394	*Protea repens* infructescense; Stellenbosch, South Africa	JX091495	JX091629	JX140707	-	-
	DTO 182-A3 = CV735 = CV0735	*Protea repens* infructescense; Stellenbosch, South Africa	JX091496	JX091630	JX140708	-	-
	DTO 182-A6 = CV787 = CV0787	Air, Malmesbury; South Africa	JX091497	JX091631	JX140709	-	-
	DTO 183-A7 = CV1426	*Protea repens* infructescense; Malmesbury, South Africa	JX091493	JX091632	JX140710	-	-
** *Talaromyces rotundus* **	CBS 369.48 (T)	Cardboard; Norway	JN899353	KJ865730	KJ885278	-	KM023275
** *Talaromyces* ** ** *ruber* **	CBS 132704 (T) = DTO193H6	Air craft fuel tank; United Kingdom	JX315662 = NR 111780	JX315629	KF741938	JX315681	JX315700
	CBS 196.88	Unknown	JX315666 = JN899312	JX315627	JX315657	JN680278 = JX315685	-
	CBS 237.93	Unknown	JX315667	JX315628	JX315656	JX315686 = JN899306	-
	CBS 370.48	Currency paper; Washington, USA	JX315673	JX315630	JX315649	JX315692	-
	CBS 868.96	Unknown	JX315677	JX315631	JX315643	JX315696 = JN899309	-
** *Talaromyces rubicundus* **	CBS 342.59 (T)	Soil; Georgia	JN899384	JX494309	KF741956	JN680301	KM023296
** *Talaromyces rugulosus* **	CBS 371.48 (T)	Roating potato tubers (*Solanum tuberosum*), USA	KF984834	KF984575 = JX494297	KF984702	JN680302	KF984925
	CBS 344.51	Unknown; Japan; Type of *Penicillium echinosporum*	KF984858	KF984574	KF984701	-	KF984924
	CBS 137366 = DTO61E8	Air sample, beer producing factory; Kaulille, Belgium; Type of *Penicillium chrysitis*	KF984850	KF984572	KF984700= JX140720	-	KF984922
	NRRL 1053	Unknown	KF984848	KF984577	KF984710	-	KF984945
	NRRL 1073	decaying twigs; France; Type of *Penicillium tardum* and *Penicillium elongatum*	KF984832	KF984579	KF984711	-	KF984927
** *Talaromyces ryukyuensis* **	NHL 2917 (T) = DTO 176-I6 = strain: NHL2917	Soil; Naha, Japan	AB176628 = NR147414	-	-	-	-
** *Talaromyces sayulitensis* **	CBS 138204 (T) = DTO 245-H1	House dust; Mexico	KJ775713	KJ775206	KJ775422	-	-
	CBS 138205 = DTO 245-H2	House dust; Mexico	KJ775714	KJ775207	KJ775423	-	-
	CBS 138206 = DTO 245-H3	House dust; Mexico	KJ775715	KJ775208	KJ775424	-	-
	NRRL 1064	Corn; Unknown	MH793034	MH792906	MH792970	-	MH793097
	NRRL 6420	Corn; Unknown	MH793041	MH792913	MH792977	-	MH793104
	FMR 15842	Unknown	-	LT898325	-	-	-
	BEOFB2600m	Unknown; Serbia	MH630050	MH780060	-	-	-
	BEOFB2601m	Unknown; Serbia	MH630051	MH780061	-	-	-
** *Talaromyces scorteus* **	CBS 340.34 (T) = NRRL 1129	Military equipment; Japan	KF984892 = NR153234 = KF196908	KF984565 = KF196851	KF984684 = KX946895	KF196953	KF984916 = KF196961
	CBS 233.60	Milled Californian rice; Japan; Type of *Talaromyces phialosporus*	KF984895	KF984562 = HQ156949	KF984683	JN680282	KF984917
	CBS 499.75	Unknown; Nigeria	KF984894	KF984563	KF984685	-	KF984918
	CBS 500.75	Unknown; Sierra Leone	KF984896	KF984564	KF984687	-	KF984919
	DTO 270-A6	House dust; Thailand	KF984893	KF984561	KF984686	-	KF984915
** *Talaromyces siamensis* **	CBS 475.88 (T)	Forest soil; Thailand	JN899385 = NR 103683.2	JX091379	KF741960	-	KM023279
	DTO 269-I3	House dust; Thailand	KJ775726	KJ775219	KJ775428	-	-
** *Talaromyces solicola* **	CBS 133445 (T) = DAOM 241015 = *Penicillium* sp. CMV-2008d isolate Pen193 = Penicillium sp. CMV-2008d isolate CV191	Soil; Malmesbury, South Africa	FJ160264	GU385731	KJ885279	-	KM023295
	CBS 133446	Soil; Malmesbury, South Africa	KF114730	KF114775	-	-	-
** *Talaromyces stipitatus* **	CBS 375.48 (T)	Decaying wood; Louisiana, USA	JN899348 = NR 147424	KM111288	KF741957	JN680303	KM023280
	NBRC 100533	Unknown	-	AB773824	-	AB856423	-
** *Talaromyces* ** ** *stollii* **	CBS 408.93 (T)	AIDS patient; the Netherlands	JX315674 = NR 111781	JX315633	JX315646	JX315693	JX315712
	CBS 169.91	Unknown substrate; South Africa	JX315664	JX315634	JX315647	JX315683	-
	CBS 265.93	Bronchoalveolar lavage of patient after lung transplantation (subclinical); France	JX315670	JX315635	JX315648	JX315689	-
	CBS 581.94	Unknown	JX315675	JX315632	JX315645	JX315694	-
	CBS 624.93	*Ananas camosus* cultivar; Martinique	JX315676	JX315636	JX315644 = JX965209	JX315695 = JX965281	JX965315
	NRRL 1768	USA, Georgia, isol ex peanut, *RJ Cole*, 1974.	-	-	-	-	MH793098
	NRRL 62122	Unknown	-	-	-	-	MH793127
	NRRL 62160	Unknown	-	-	-	-	MH793136
	NRRL 62163	Unknown	-	-	-	-	MH793137
	NRRL 62165	Soil; Unknown	-	-	-	-	MH793138
	NRRL 62171	Unknown	-	-	-	-	MH793140
	NRRL 62227	Corn; Unknown	-	-	-	-	MH793144
** *Talaromyces subinflatus* **	CBS 652.95 (T)	Copse soil; Japan	JN899397 = NR 137080	KJ865737 = JX494288	KJ885280	JN899301	KM023308
** *Talaromyces tardifaciens* **	CBS 250.94 (T)	Paddy soil; Bhaktapur, Nepal	JN899361	KC202954 = KF984560	KF984682	JN680283	KF984908
** *Talaromyces thailandensis* **	CBS 133147 (T)	Soil; Thailand	JX898041 = NR 147428	JX494294	KF741940	JX898043	KM023307
** *Talaromyces trachyspermus* **	CBS 373.48 (T)	Unknown; USA	JN899354 = NR 147425	KF114803	KJ885281	JN121664	JF417432
	CBS 116556	Pasteurised canned strawberries; Germany	KM066170	KM066126	MK451694	-	-
	CBS 118437	Soil; Marocco	KM066169	KM066127	MK451695	-	-
	CBS 118438	Soil; Marocco	KM066166	KM066128	MK451696	-	-
** *Talaromyces tratensis* **	CBS 113146 (T) = CBS 133146 (*RPB1*)?	Soil; Trat, Thailand	KF984891	KF984559	KF984690	JX898042	KF984911
	CBS 137400 = DTO 270-F5	House dust; Mexico	KF984889	KF984557	KF984688	-	KF984909
	CBS 137401 = NRRL1013	Carbonated beverage; Washington D.C., USA	KF984890	KF984558	KF984689	-	KF984910
** *Talaromyces tumuli* **	NRRL 62151 (T)	Soil; Unknown	MH793071= NR 165528	MH792944	MH793008	-	MH793135
	NRRL 6013	Unknown	MH793038	MH792910	MH792974	-	MH793101
	NRRL 62469	Peanut; Unknown	MH793089	MH792962	MH793026	-	MH793153
	NRRL 62471	Peanut; Unknown	MH793090	MH792963	MH793027	-	MH793154
	F-3	Unknown	MT434004	-	-	-	-
** *Talaromyces ucrainicus* **	CBS 162.67 (T)	Unknown	JN899394 = NR 153205	KF114771	KJ885282	JN680277	KM023289
	CBS 127.64	soil treated with cyanamide; Germany; Type of *Talaromyces ohiensis*	KM066173	KF114772	-	JN680272	-
	CBS 583.72A	Soil; Japan	KM066174	KM066130	-	-	-
	CBS 583.72C	Soil; Japan	KM066175	KM066131	-	-	-
** *Talaromyces udagawae* **	CBS 579.72 (T)	Soil; Misugimura, Japan	JN899350 = NR 145156	KF114796	KX961260	JN680310	-
** *Talaromyces unicus* **	CBS 100535 (T)	Soil; Taiwan	JN899336 = NR 157429	KJ865735	KJ885283	JN680324	-
** *Talaromyces varians* **	CBS 386.48 (T)	Cotton yarn; England	JN899368 = NR 111689	KJ865731	KJ885284	JN680305	KM023274
** *Talaromyces veerkampii* **	CBS 500.78 (T)	Unknown	KF741984 = NR 153228	KF741918	KF741961	-	KX961279
	NRRL 6095	Unknown	MH793040	MH792912	MH792976	-	MH793103
	NRRL 62286	Wheat flour; Unknown	MH793085	MH792958	MH793022	-	MH793149
	IBT18366	Unknown	KF741993	KF741924	KF741973	-	-
	CMV005D6	Soil; South Africa	MK450751	MK451043	-	-	-
** *Talaromyces verruculosus* **	NRRL 1050 (T) = CBS 388.48	Soil; Texas, USA	KF741994	KF741928	KF741974	-	KM023306
	CBS 254.56	Unknown; Yangambi, Zaire	KF741980	KF741913	KF741944	-	-
	DTO 129-H4	House dust; Thailand	KJ775698	KJ775191	KJ775419	-	-
	DTO 129-H5	House dust; Thailand	KJ775699	KJ775192	KJ775420	-	-
	AX2101 I	Metallic surface; Para, Brazil	KJ413368	KJ413340	-	-	KJ476428
** *Talaromyces* ** ** *viridis* **	CBS 114.72 (T) = *Sagenoma viride*	Soil; Australia	AF285782 = MH860406 = NR160136	JX494310	KF741935	JN121571	JN121430
** *Talaromyces viridulus* **	CBS 252.87 (T)	Soil from bank of creek floading into Little river; New South Wales	JN899314 = NR103663.2	JX091385	KF741943	JN680284 = JN121620	JF417422
** *Talaromyces wortmannii* **	CBS 391.48 (T)	Soil; Denmark	KF984829	KF984648	KF984756	JN121669	KF984977 = JF417433
	CBS 319.63	Unknown	KF984828	KF984651	KF984755	-	KF984961
	CBS 385.48 = NRRL 1048	coconut matting; Johannesburg, South Africa; Type of *Talaromyces variabilis*	KF196915	KF196853 = JX494295	KF196878	JN680304	KF196975 = KX657552
	CBS 895.73	Unkown; Japan	KF984811	KF984626	KF984737	-	KF984982
	CBS 137376 = DTO 176-I7	soil; Japan; Type of *Talaromyces sublevisporus*	KF984800	KF984632	KF984724	-	KF984979
	NRRL 2125 = DTO 278-E7	Weathering canvas; Panama	KF984797	KF984635	KF984731	-	KF984991
** *Talaromyces xishaensis* **	HMAS 248732 (T)	China	NR147445	-	-	-	-
	-	China	KU644580	KU644581	KU644582	-	-
** *Talaromyces yelensis* **	CBS 138210 (T) = DTO 268-E5	House dust; Micronesia	KJ775717	KJ775210	KP119162	-	KP119164
	CBS 138209 = DTO 268-E7	House dust; Micronesia	KJ775719 = NR 145183	KJ775212	KP119161	-	KP119163

The result of NCBI standard nucleotide blast is considered preferentially; moreover, the aim of adding type strains genus *Talaromyces* is to make the phylogenetic tree more plentiful. Genus and species in the columns are represented by bold Italic. T indicates ex type. Sect. *Talaromyces*; sect. *Helici*; sect. *Purpurei*; sect. *Trachyspermi*; sect. *Bacillispori*; sect. *Subinflati*; sect. *Islandici*.

**Table 2 jof-08-00155-t002:** Morphological comparison of strain JP-NJ4 with *Talaromyces liani* and *Talaromyces brevis*.

Morphological Characters			Species		
			*T. liani* (Yilmaz et al., 2014)	*Talaromyces* Strain JP-NJ4	*T. brevis* (Sun et al., 2020)
**Macromorphological Characters**	**Ascomata**		Present after 25 °C, 7 d on OA and MEA (at 30 °C abundant yellow ascomata)	Present after 25 °C, 7 d on OA, 25 °C, 14 d on CZ, and 30 °C, 14 d on CYA and MEA	Present after 25 °C, 7 d on OA
	**Growth rate (mm) Diam (diameter), 7 d**	CZ (25 °C)	Unknown	29–33	Unknown
		CYA (25 °C)	20–30	25–29	30–31
		CYA (30 °C)	25–37	30–37	28–30
		CYA (37 °C)	20–25	21–31	25–26
		MEA (25 °C)	35–45	31–33	50–51
		MEA (30 °C)	50–55	35–41	57–60
		MEAbl (25 °C)	Unknown	34–43	Unknown
		OA (25 °C)	35–40	38–44	39–43
		DG18 (25 °C)	10–17	15–18	13–15
		CYAS (25 °C)	No growth	No growth	No growth
		YES (25 °C)	35–40	30–40	42–43
		CREA (25 °C)	10–20	18–24	13–14
		HAY (25 °C)	Unknown	No growth	Unknown
	Colour of CYA reverse		Light orange and light yellow (5A5–4A5)	Centre pastel yellow (2D4) to pale yellow (1A4)	Ochreous (44)
	**Soluble pigment**		Absent on CYA (in some isolates yellow) and MEA at 25 °C, 7 d	Weak yellow and orange soluble pigments present on CYA and MEA at 25 °C, 7 d; Strong red soluble pigments present on MEA at 25 °C, 14 d	Absent
	**MEA colony texture**		Velvety and floccose	Velvety to floccose	Floccose
	**Acid production on CREA**		Absent (in some isolates very weak)	Present strong	Present
**Micromorphological Characters**	**Conidiophore**		Present	Present	Present
	**Conidiophore branching**		Mono- to biverticillate	Monoverticillate to biverticillate, reduced conidiophores consisting of solitary phialides	Mono- to biverticillate
	**Conidium**	**Shape**	Ellipsoidal	Globose to subglobose; (sometimes ovoid)	Subglobose to fusiform
		**Size (μm)**	2.5–4(–4.5) × 2–3.5	2–3 × 3; (3 × 3–3.5)	3–4(–5) × 2.5–3.5(–4.5)
		**Ornamentation**	Smooth	Smooth	Smooth
	**Ascoma colour**		Yellow to orange red	Yellow	Yellow to orange
	**Ascoma shape**		Globose to subglobose	Globose to subglobose	Globose to subglobose
	**Ascoma size (μm)**		150–550 × 150–545	300–950 × 300–1000	400–550 × 400–550
	**Asci size (μm)**		9–13 × 7.5–11	10–12 × 8–10	Unknown
	**Ascospore**	**Shape**	Broadly ellipsoidal	Broadly ellipsoidal	Ellipsoidal
		**Size (μm)**	4–6 × 2.5–4	3.5–5 × 2–3	3.5–4.5 × 3–4
		**Ridges**	Absent	Absent	Absent
		**Ornamentation**	Spiny	Spiny	Spiny

## Data Availability

The materials are available as Appendix A (Appendix A). Publicly available datasets were analyzed in this study. This data can be found here: https://www.ncbi.nlm.nih.gov/genbank, accessed on 18 January 2022; accession number ITS = MW130720, *BenA* = MW147759, *CaM* = MW147760, *RPB1* = MW147761, *RPB2* = MW147762.

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
