# Peer review of "Fine Identification and Classification of a Novel Beneficial Talaromyces Fungal Species from Masson Pine Rhizosphere Soil"

_jof, 2022, doi:10.3390/jof8020155_

Round 1

Reviewer 1 Report

I found that authors implemented all the suggested corrections and modifications in revised manuscript. 

Author Response

Thank you for your professional suggestions, as well as your affirmation and support of our manuscript. I wish you a happy life, good health and everything goes well with you.

Reviewer 2 Report

Comments from the last round of reviews have been largely addressed. Just one minor comment:

Line 98: With the renaming of Penicillium marneffei to Talaromyces marneffei, peniciliosis has been renamed as talaromycosis, too.

Author Response

Point 1: Line 98: With the renaming of Penicillium marneffei to Talaromyces marneffei, peniciliosis has been renamed as talaromycosis, too.

Response 1: Thank you for your affirmation, help and support of our manuscript. Thanks for your professional advice, we have revised peniciliosis in the manuscript to talaromycosis.

I wish you a happy life, good health and everything goes well with you

This manuscript is a resubmission of an earlier submission. The following is a list of the peer review reports and author responses from that submission.

Round 1

Reviewer 1 Report

Dear Author,

  1. Line numbers is missing in the manuscript due to which preparing comments is bit difficult. 
  2. Introduction is too much lengthy. reduce the size with precise information.
  3.  Please see the attached file for more comments. 

Reviewer 2 Report

The manuscript by Sun et al. described a novel Talaromyces species, T. nanjingensis, which was previously isolated from the rhizosphere soil of Pinus massoniana in Nanjing. This unique strain was characterised comprehensively for the determination of its taxonomic position and it was found to be the most closely related to, but distinct from, T. liani and T. brevis. Results presented in the manuscript support this proposal.

Below are some comments which the authors should consider to address for improvement.

1. The manuscript is rather lengthy. There are a lot of unrelated materials. For example, p. 30 first paragraph & p.35 first and second paragraphs. These are not the focus of this manuscript and so can be removed. Tables 1 & 4 can be moved as supplementary materials whereas Tables 2 & 3 can be removed. Figures 7, 8, 9 & 10 can also be moved as supplementary materials. Too many of these in the main text would distract/confuse the readers. The manuscript as a whole should also be made more concise.

2. Keywords: Please refer to Schoch CK et al. 2017. Using standard keywords in publications to facilitate updates of new fungal taxonomic names. IMA Fungus. 8:A70-A73. for the use of keywords for publications describing novel fungal taxa.

3. Please also refer to Catherine Aime M. et al. 2021. How to publish a new fungal species, or name, version 3.0. IMA Fungus. 12:11. for the requirements for the description of novel fungal taxa.

4. How did the authors prepare the holotype specimen? Was it a dried culture or preserved in lyophilised state?

5. If the holotype was preserved in a metabolically inactive state such as cryopreservation, lyophilisation or as dried cultures, this must be stated in the protologues according to Chapter F of the International Code of Nomenclature for algae, fungi and plants (May T.W. et al. 2019. Chapter F of the International Code of Nomenclature for algae, fungi, and plants as approved by the 11th International Mycological Congress, San Juan, Puerto Rico, July 2018. IMA Fungus. 10:21.)

6. Ideally duplicate living cultures prepared from the holotype strain should be deposited to culture collections in more than one country (Catherine Aime M. et al. 2021. How to publish a new fungal species, or name, version 3.0. IMA Fungus. 12:11.)

7. Introduction to Talaromyces section Talaromyces is missing in the Introduction.

8. What is the purpose of having both Tables S1 and S2? What are the differences between them? Only one of them may be necessary and should be moved to the main text.

9. Section 2.3: 'All new sequences for phylogenetic analysis...' Were only sequences for strain JP-NJ4 sequenced by the authors? Were all the other sequences used for phylogenetic analysis retrieved from GenBank? This sentence is confusing and may be misleading.

10. Section 3.2: The first paragraph is irrelevant. A number of the GenBank entries were incorrectly annotated. It is not meaningful to merely report the BLAST findings. The second paragraph was not part of the 'Results'

11. Too many trees are shown in the main text. It may only be necessary to show the 5-loci tree in the main text and all the others can be moved as supplementary materials. 

12. Based on the phylogenetic analysis, it appears that strain JP-NJ4 is the most closely related to, but distinct from, T. liani and T. brevis. For T. brevis, RPB1 sequence is not available for any of the 3 strains. It may be beneficial for the authors to at least obtain the ex-type strain for T. brevis and sequence its RBP1 sequence so as to include this species in the 5-loci tree. Otherwise, this tree may not be useful to infer the phylogenetic position of strain JP-NJ4 at all. Similarly, the authors should also consider to obtain the strain 'Talaromcyes liani' KUC21412. Only benA sequence is available for this strain but it appears that the benA sequence for this strain is highly similar to JP-NJ4. This strain is available at the KUC Culture Collection and sequencing of additional loci could help determine whether these two strains belong to the same novel species. It would be beneficial to propose a novel species with multiple strains being characterised. It is not meaningful to show the small trees 'WITHOUT' T. brevis or KUC21412. These strains are important to ascertain the phylogenetic position of strain JP-NJ4.

13. If strain KUC21412 is found to belong to the same novel species as JP-NJ4, the authors may need to consider an alternative species epithet for this novel species since KUC21412 was not isolated from Nanjing?

14. Strain KUC21412 should be described as 'T. liani' (with quotation marks) since its current species identity is doubtful.

15. p.11 First paragraph 'BenA is the secondary barcode with the highest reliability ..... ' Please provide references for this statement.

16. Phylogenetic trees: Please include the 'T' (ex-type strain) for T. nanjingensis JP-NJ4.

17. p.24-25: the explanation for including these images in the manuscript is not necessary.

18. p.26 'The macromorphological differences of T. nanjingensis strain JP-NJ4, T. liani and T. brevis on various media were obvious'. Please elaborate. What are the differences?

19. p.30 First paragraph: the first half of this paragraph is unnecessary.

20. Table 5: It would be better for the authors to obtain at least the ex-type strains for the comparator species and characterise the morphological features themselves. Citing morphological characters for comparison is not a good practice since these morphological characters can differ under different laboratory conditions.